# FG-CLIP 2: A Bilingual Fine-grained Vision-Language Alignment Model

**Chunyu Xie** [1 2 *]  **Bin Wang** [2 *]  **Fanjing Kong** [2]  **Jincheng Li** [2]  **Dawei Liang** [2]  **Ji Ao** [2]
**Dawei Leng** [2]  **Yuhui Yin** [2]

Homepage: https://360cvgroup.github.io/FG-CLIP
Code: https://github.com/360CVGroup/FG-CLIP
Model&Dataset: https://huggingface.co/collections/qihoo360/fg-clip-2

## Abstract

Fine-grained vision-language understanding requires precise alignment between visual content and linguistic descriptions, a capability that remains limited in current models, particularly in non-English settings. While models like CLIP perform well on global alignment, they often struggle to capture fine-grained details in object attributes, spatial relations, and linguistic expressions, with limited support for bilingual comprehension. To address these challenges, we introduce FG-CLIP 2, a bilingual vision-language model designed to advance fine-grained alignment for both English and Chinese. Our approach leverages rich fine-grained supervision, including region-text matching and long-caption modeling, alongside multiple discriminative objectives. We further introduce the Textual Intra-modal Contrastive (TIC) loss to better distinguish semantically similar captions. Trained on a carefully curated mixture of large-scale English and Chinese data, including a newly released 12M Chinese region-text dataset, FG-CLIP 2 achieves powerful bilingual performance. To enable rigorous evaluation, we present a new benchmark for Chinese multimodal understanding, featuring long-caption retrieval and bounding box classification. Extensive experiments on 29 datasets across 8 tasks show that FG-CLIP 2 outperforms existing methods, achieving state-of-the-art results in both languages. We release the model, code, and benchmark to facilitate future research on bilingual fine-grained vision-language alignment.

[1]Institute of Unmanned System, Beihang University [2]360 AI Research. Correspondence to: Dawei Leng <lengdawei@360.cn>.

*Proceedings of the $43^{rd}$ International Conference on Machine Learning*, Seoul, South Korea. PMLR 306, 2026. Copyright 2026 by the author(s).

## 1. Introduction

Vision-language alignment models (Tschannen et al., 2025; Chuang et al., 2025) have undergone rapid evolution in recent years, driven by pioneering works such as CLIP (Radford et al., 2021), which introduced large-scale contrastive pre-training on image-text pairs and demonstrated remarkable success in learning joint multimodal representations. These models excel at global alignment tasks such as zero-shot image classification and image-text retrieval, forming the foundation for a wide range of multimodal understanding systems (Zhu et al., 2025; Team et al., 2025a; Li et al., 2025a; Wu et al., 2025; Team et al., 2025b). Their ability to align visual and linguistic concepts without explicit supervision has enabled strong generalization to diverse scenarios, including visual question answering (Lu et al., 2025; Wang et al., 2025a), image captioning (Bai et al., 2025; Li et al., 2025b), and content-based retrieval (Zhang et al., 2024a; Wei et al., 2024). However, their performance often degrades on fine-grained understanding tasks that require discriminating between similar object attributes, spatial configurations, or semantic distinctions. Such tasks demand precise alignment at both visual and linguistic levels: visually, they involve recognizing objects, attributes, and their spatial arrangements; linguistically, they require distinguishing between semantically similar expressions. This performance gap stems from reliance on coarse-grained image-text pairs during training, which encourages thematic alignment while failing to capture the fine-grained correspondences essential for robust visual grounding or attribute recognition.

Several recent works have sought to address these limitations. Approaches such as FineCLIP (Jing et al., 2024) and LongCLIP (Zhang et al., 2024a) improve fine-grained understanding by incorporating region-level signals or supporting longer textual inputs. FG-CLIP (Xie et al., 2025) further advances fine-grained discrimination through large-scale data curation and attribute-aware hard negative sampling. However, achieving robust discrimination between highly similar textual descriptions or subtle visual variations remains challenging, suggesting opportunities for more advanced

training objectives. Moreover, prior fine-grained approaches are predominantly developed and evaluated in English, leaving Chinese–English bilingual scenarios relatively underexplored. Meanwhile, models designed for Chinese, such as Chinese-CLIP (Yang et al., 2022) and R2D2 (Xie et al., 2023), primarily focus on coarse-grained alignment and lack fine-grained modeling capability. Therefore, extending fine-grained alignment to bilingual vision–language settings remains an important yet underexplored problem.

To address these challenges, we propose FG-CLIP 2, a unified framework for bilingual fine-grained vision-language alignment. Our training strategy employs a two-stage paradigm to progressively refine model capabilities. In the first stage, we perform initial global alignment with both short and long textual descriptions to capture coarse and detailed semantic content at the early phase of training. In the second stage, we incorporate fine-grained learning objectives that improve regional alignment, discriminative capability, and cross-modal ranking performance. To further refine the model's ability to distinguish similar region-level descriptions, we propose the Textual Intra-modal Contrastive (TIC) loss, which learns from filtered hard negatives among high-similarity text pairs. FG-CLIP 2 is trained on large-scale, high-quality bilingual datasets with careful curation, enabling strong performance in both English and Chinese across diverse fine-grained vision-language tasks.

We further contribute a large-scale Chinese region-text dataset, FineRegion-CN, containing 12 million images with fine-grained region descriptions to fill the gap of Chinese region-level training data. Additionally, we introduce a new benchmark suite to advance evaluation in Chinese multimodal understanding, featuring challenging tasks such as long caption image-text retrieval and bounding box classification in Chinese that go beyond conventional short-text retrieval and assess fine-grained comprehension more rigorously. Extensive experiments show that FG-CLIP 2 outperforms existing models on 29 datasets across 8 vision-language tasks in both Chinese and English, demonstrating powerful bilingual fine-grained vision-language alignment capability. To support future research and real-world deployment, our model, training data, code, and benchmark are made publicly available.

## 2. Related Work

Vision-language alignment models trained on large-scale data, such as CLIP (Radford et al., 2021), EVA-CLIP (Sun et al., 2023), SigLIP (Zhai et al., 2023), MetaCLIP (Xu et al., 2024) and DFN (Fang et al., 2024) have demonstrated strong zero-shot capabilities but primarily focus on global semantic alignment and are often trained on English-only corpora. While these models serve as backbones for downstream tasks including multimodal reasoning (Bai et al.,

2025), open-vocabulary detection (Fu et al., 2025), and segmentation (Cho et al., 2024; Wang et al., 2025b), they lack fine-grained and multilingual understanding, limiting their broader applicability.

Recent efforts aim to improve localization capability and dense feature alignment through region-level supervision. Methods like AlphaCLIP (Sun et al., 2024), UMG-CLIP (Shi et al., 2024), FineCLIP (Jing et al., 2024), and FG-CLIP (Xie et al., 2025) leverage bounding boxes or masked regions to enhance local correspondence, while CLOC (Chen et al., 2024a), TIPS (Maninis et al., 2025), and SigLIP 2 (Tschannen et al., 2025) introduce architectural or training enhancements for richer feature generation. On the multilingual front, Chinese-CLIP (Yang et al., 2022) and R2D2 (Xie et al., 2023) target Chinese understanding, and MetaCLIP 2 (Chuang et al., 2025) scales multilingual data collection for broader language coverage. However, these works often treat fine-grained and multilingual understanding separately, and none explicitly optimize both in a unified framework. This gap inspires our work.

## 3. Approach

Figure 1 illustrates FG-CLIP 2, a vision-language model supporting Chinese and English. Our approach follows a two-stage hierarchical learning framework: the first stage establishes strong semantic alignment by training on large-scale image-text pairs, each associated with both a short caption and a long caption; the second stage extends this learning by incorporating region-level alignment and fine-grained contrastive signals, enabling the model to preserve holistic scene understanding while enhancing its ability to discriminate fine-grained visual-language correspondences in both languages.

### 3.1. Architecture

We build upon the SigLIP 2 (Tschannen et al., 2025) dual-encoder framework, introducing key adaptations for fine-grained understanding and bilingual alignment. For the text encoder, we extend the maximum input length from 64 to 196 tokens to accommodate longer descriptions. On the vision side, we adopt a data-adaptive resolution strategy: the target resolution is selected from {128, 256, 576, 784, 1024} based on the maximum image size per mini-batch, avoiding the stochastic sampling of SigLIP 2 and ensuring consistent training and inference behavior with minimal upscaling or downscaling. Global vision features are aggregated via a masked attention pooling head (Zhai et al., 2022), while region embeddings are generated through an additional self-attention module. This design enables fine-grained, sub-image alignment and circumvents the information bottleneck caused by CLS-only pooling.

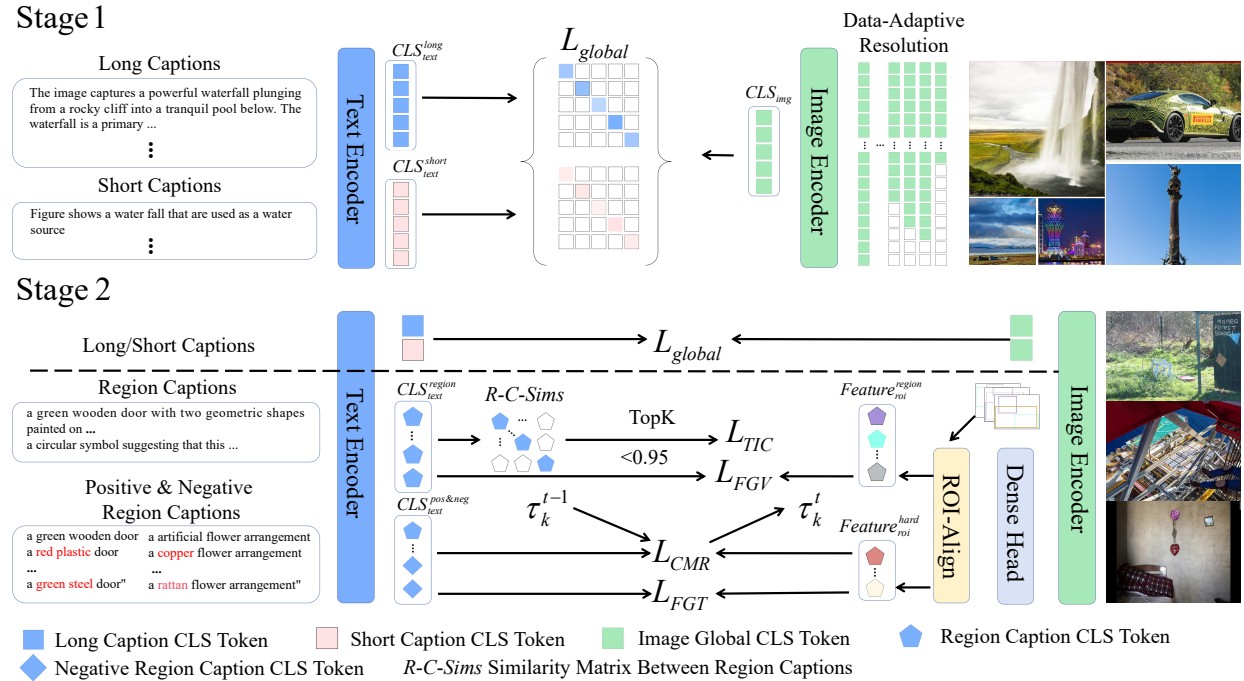

*Figure 1.* Overview of FG-CLIP 2. The framework consists of a two-stage training pipeline. Stage I performs bilingual global alignment using data-adaptive image resolution, matching image global tokens with long and short caption embeddings via $\mathcal{L}_{\text{Global}}$. Stage II introduces fine-grained region-caption supervision alongside the global alignment. It employs ROI features and explicit hard-negative mining to compute auxiliary objectives: Fine-Grained Visual Learning ($\mathcal{L}_{\text{FGV}}$), Fine-Grained Textual Learning ($\mathcal{L}_{\text{FGT}}$), Textual Intra-modal Contrastive loss ($\mathcal{L}_{\text{TIC}}$), and Cross-modal Rank Loss ($\mathcal{L}_{\text{CMR}}$), thereby enhancing discrimination for semantic details.

## 3.2. Training Objectives

Our training proceeds in two stages. Stage I focuses on Global Alignment Learning. In Stage II, this objective is jointly optimized with four additional learning objectives, detailed in the subsequent paragraphs.

**Global Alignment Learning.** We adopt the sigmoid loss from SigLIP (Zhai et al., 2023), which treats image-text matching as a binary classification. For each image-text pair, similarity scores are computed across all pairs in the batch, and logistic regression is applied to distinguish positive from negative pairs. To enrich textual supervision, we include both original short captions and long captions generated by Large Multimodal Models (LMMs). This dual-caption strategy provides complementary signals: concise labels for global semantics and detailed descriptions for richer context. We denote this global alignment loss as $\mathcal{L}_{\text{Global}}$.

**Fine-Grained Visual Learning.** For each annotated region, we extract patch-level embeddings and apply RoIAlign (He et al., 2017) to obtain region-specific features. Corresponding text embeddings are derived from descriptions aligned to each bounding box. The regional contrastive loss encourages alignment between matched region-text pairs, promoting fine-grained cross-modal understanding. This loss term is denoted as $\mathcal{L}_{\text{FGV}}$.

**Fine-Grained Textual Learning.** Following FG-CLIP (Xie et al., 2025), we leverage hard negatives from the Fine-HARD dataset. Each positive region-text pair is paired with 10 semantically similar negatives, constructed by perturbing key attributes (e.g., color, count, action) while preserving syntactic structure. The model is trained with a binary classification loss over the 1 positive and 10 negatives, encouraging the model to assign higher scores to the matched pair. This objective enhances the model's ability to discriminate subtle textual differences. We formulate this discriminative objective as $\mathcal{L}_{\text{FGT}}$.

**Textual Intra-modal Contrastive Loss.** While cross-modal alignment ensures image-text correspondence, the text encoder itself often lacks sufficient discriminative pressure to separate semantically similar but distinct region descriptions, which is critical for fine-grained visual grounding. To address this, we propose the Textual Intra-modal Contrastive (TIC) loss, which operates purely within the text modality to sharpen the text encoder's representation space. Given a batch of region texts, we compute pairwise similarities and filter out pairs with sim > 0.95 to avoid over-penalization. The top-10 most similar texts per sample are selected as hard negatives. The TIC loss is then defined as:

$$\mathcal{L}_{\text{TIC}} = -\sum_{i=1}^{N} \log \frac{1}{\sum_{T_m \in \mathcal{T}_i} \exp(S(T_i, T_m))}, \quad (1)$$

where $\mathcal{T}_i$ denotes the set of filtered hard negatives for text $T_i$. This encourages the text encoder to assign lower similarity to hard negative pairs, thereby improving its ability to separate semantically close but distinct region descriptions.

**Cross-modal Rank Loss with Global Threshold Synchronization.** We adopt the Cross-modal Rank (CMR) loss (Zhang et al., 2024b) to enforce a margin between positive and hard negative pairs, thereby strengthening the model's discrimination of semantic boundaries. For a positive pair $(I, T)$ and hard negative $T_k$, the loss is defined as:

$$\mathcal{L}_{\text{CMR}} = \max\left(0, S(I, T_k) - S(I, T) + \tau_k\right), \quad (2)$$

where $S(\cdot, \cdot)$ denotes cosine similarity. At training step $t$, the margin $\tau_k$ is synchronized globally across all GPUs via all-reduce, where $\mathcal{B}_{\text{global}}$ denotes the union of all local batches across GPUs:

$$\tau_k^t = \frac{1}{|\mathcal{B}_{\text{global}}|} \sum_{(I,T) \in \mathcal{B}_{\text{global}}} \left(S^{t-1}(I, T) - S^{t-1}(I, T_k)\right), \quad (3)$$

with $S^{t-1}(\cdot, \cdot)$ denoting similarities computed at the previous step. This design ensures stable and consistent thresholding in distributed training.

In Stage II, the training objective combines these terms as:

$$\mathcal{L} = \mathcal{L}_{\text{Global}} + \lambda_1 \mathcal{L}_{\text{FGV}} + \lambda_2 \mathcal{L}_{\text{FGT}} + \lambda_3 \mathcal{L}_{\text{TIC}} + \lambda_4 \mathcal{L}_{\text{CMR}}, \quad (4)$$

with fixed weights $\lambda_1 = 0.1$, $\lambda_2 = 0.5$, $\lambda_3 = 0.1$, $\lambda_4 = 0.4$, chosen to ensure stable and effective multi-objective optimization. The selection strategy for loss weights is detailed in Appendix F.

### 3.3. Training Data

In the first stage, we train on image-text pairs from diverse sources, with a particular emphasis on enhancing semantic depth and linguistic coherence. For English, we adopt an enhanced version of the LAION-2B dataset (Xie et al., 2025), which augments the original short captions with detailed long captions generated by LMMs. The original captions are often fragmented and contain keyword stacking or irrelevant noise, making them insufficient for training models to understand rich, compositional language. We retain original captions to preserve diversity of natural language expressions, while simultaneously training on their long-caption counterparts. This dual-caption strategy enables the model to learn from both concise, real-world descriptions and semantically dense, contextually coherent narratives. For Chinese, we combine three datasets: Wukong (100M pairs) (Gu et al., 2022), Zero (250M pairs) (Xie et al., 2023), and a large-scale in-house dataset (500M pairs).

In the second stage, we extend training with fine-grained region-text pairs to further improve spatial grounding. For English, we use the FineHARD dataset (Xie et al., 2025), which includes 12 million images, 40 million bounding boxes with fine-grained region descriptions, and 10 million hard negative samples. For Chinese, we construct a complementary dataset, termed FineRegion-CN, following the region-text annotation pipeline of FineHARD. It contains 12 million images with bounding boxes and fine-grained region descriptions in Chinese. FineRegion-CN are publicly released to support research on Chinese fine-grained multimodal understanding. An overview of our training datasets is provided in Appendix A.

### 3.4. Bilingual Evaluation Protocols

Existing multimodal benchmarks in English are diverse and well-established, covering a broad spectrum of vision-language tasks such as fine-grained object-level understanding (e.g., FG-OVD (Bianchi et al., 2024)), open-vocabulary object detection (e.g., LVIS (Gupta et al., 2019), COCO (Lin et al., 2014)), and image-text retrieval (e.g., Flickr30K (Young et al., 2014), DCI(Urbanek et al., 2024)). These resources enable comprehensive evaluation of model capabilities across different granularities and semantic complexities. In contrast, Chinese multimodal datasets remain limited in scope and diversity, with most focusing on short-caption retrieval tasks such as COCO-CN (Li et al., 2019) and Flickr30k-CNA (Xie et al., 2023). Such benchmarks are insufficient for evaluating fine-grained cross-modal alignment, particularly at the region level or with long, descriptive textual inputs.

To address this gap, we introduce a suite of Chinese evaluation benchmarks tailored for fine-grained vision-language tasks. We first construct three long-caption image-text retrieval datasets: LIT-CN, DCI-CN, and DOCCI-CN. These datasets support the evaluation of cross-modal alignment with rich and descriptive textual inputs. We then present BoxClass-CN, a region-based classification dataset designed to assess region-level vision-language alignment in Chinese.

LIT-CN integrates diverse sources: 15,000 images from AI-Challenger Caption (Wu et al., 2019), 3,230 from MUGE (Lin et al., 2021), and 20,000 from curated web images. All images are uniformly re-captioned using Qwen2.5-VL-32B-Instruct-AWQ (Bai et al., 2025), prompted to generate rich, context-aware descriptions with an average length of 131 tokens. Images below 256×256 resolution are filtered, resulting in 33,010 high-quality image-text pairs. DCI-CN is derived from the Densely Captioned Images (DCI) dataset (Urbanek et al., 2024), with English captions translated into Chinese using the same LMM. The translations are validated by native speakers to ensure linguistic fluency and alignment with the original semantics. Similarly, DOCCI-CN is constructed from the DOCCI (Onoe et al., 2024) dataset, following an identical translation and

validation pipeline.

BoxClass-CN is a region classification dataset that evaluates the alignment between image regions and their corresponding Chinese textual descriptions. It complements existing Chinese benchmarks by providing region-level supervision and serves as an evaluation suite for assessing models' fine-grained understanding of visual content. We construct this dataset through a scalable automated pipeline based on the LAION-2B corpus (Schuhmann et al., 2022). We first sample 200,000 images and generate detailed captions using a LMM (Hong et al., 2024). These captions are parsed to extract referring expressions, which are then localized using a pretrained object detector (Cheng et al., 2024) to produce bounding box proposals. Non-maximum suppression removes overlapping boxes, and only those with region-text similarity above 0.15 (computed by FG-CLIP (Xie et al., 2025)) are retained. Detected categories undergo semantic clustering and merging, resulting in 566 semantically refined categories. These categories are translated into Chinese, and the final dataset consists of 24,221 images and 66,258 high-quality region-text pairs. We provide examples of the proposed datasets in Appendix I and J. Together, these datasets provide a rigorous and comprehensive assessment for bilingual vision-language alignment models, supporting deeper evaluation of fine-grained understanding capability.

## 4. Experiments

### 4.1. Implementation Details

The first stage is conducted on 160×ASCEND 910B NPUs, and the second stage uses 16×NVIDIA H800 GPUs. We use three vision encoder configurations: ViT-B/16, ViT-L/16, and ViT-So/16, initialized with SigLIP 2 (Tschannen et al., 2025) pre-trained weights. We employ the AdamW optimizer with a learning rate of $1\times10^{-6}$ and a weight decay coefficient of 0.001. The momentum parameters $\beta_1$ and $\beta_2$ are set to 0.9 and 0.98, respectively. A learning rate warmup strategy is applied during the first 300 iterations for stability. To accelerate training, we employ Zero-2 (Rajbhandari et al., 2020), CUDA TF32 precision, FlashAttention (Dao, 2023), and BFloat16 mixed-precision training. Batch sizes are set based on model size and training stage. In the first stage, the global batch sizes are 61,440 for ViT-B, 30,720 for ViT-L, and 18,432 for ViT-So. In the second stage, they are reduced to 4,096, 3,072, and 2,560, respectively. All models are trained for one epoch per stage.

### 4.2. Localization tasks

#### 4.2.1. FINE-GRAINED UNDERSTANDING

We evaluate open-source image-text alignment models on FG-OVD (Bianchi et al., 2024), a fine-grained benchmark emphasizing grounding in specific local image regions.

*Table 1.* Performance comparison on fine-grained understanding tasks using Top-1 accuracy.

| Method | Backbone | Fine-Grained Understanding | | | |
| | | Hard | Medium | Easy | Trivial |
| --- | --- | --- | --- | --- | --- |
| CLIP | ViT-B/16 | 12.0 | 23.1 | 22.2 | 58.5 |
| EVA-CLIP | ViT-B/16 | 14.0 | 30.1 | 29.4 | 58.3 |
| Long-CLIP | ViT-B/16 | 9.2 | 18.4 | 16.2 | 51.8 |
| FineCLIP | ViT-B/16 | 26.8 | 49.8 | 50.4 | 71.9 |
| SigLIP 2 | ViT-B/16 | 24.9 | 46.5 | 48.7 | 85.0 |
| FG-CLIP | ViT-B/16 | 46.1 | 66.6 | 68.7 | 83.4 |
| FG-CLIP 2 | ViT-B/16 | **52.3** | **76.3** | **80.3** | **92.0** |
| CLIP | ViT-L/14 | 15.4 | 25.3 | 25.7 | 38.8 |
| EVA-CLIP | ViT-L/14 | 18.3 | 38.4 | 35.2 | 62.7 |
| Long-CLIP | ViT-L/14 | 9.6 | 19.7 | 16.0 | 39.8 |
| FineCLIP | ViT-L/14 | 22.8 | 46.0 | 46.0 | 73.6 |
| SigLIP 2 | ViT-L/16 | 24.1 | 47.1 | 47.4 | 84.1 |
| FG-CLIP | ViT-L/14 | 48.4 | 69.5 | 71.2 | 89.7 |
| FG-CLIP 2 | ViT-L/16 | **55.5** | **77.5** | **83.1** | **92.5** |
| Meta CLIP 2 | ViT-H/14 | 16.5 | 36.6 | 34.7 | 79.6 |
| SigLIP 2 | ViT-So/16 | 26.0 | 48.7 | 49.9 | 87.4 |
| FG-CLIP 2 | ViT-So/16 | **54.0** | **77.4** | **79.8** | **93.5** |

*Table 2.* Performance comparison on bounding box classification tasks using Top-1 accuracy.

| Method | Backbone | COCO[80] | LVIS[1203] | BoxClass-CN[566] |
| --- | --- | --- | --- | --- |
| CLIP | ViT-B/16 | 44.2 | 20.9 | - |
| EVA-CLIP | ViT-B/16 | 30.6 | 14.4 | - |
| Long-CLIP | ViT-B/16 | 36.7 | 18.2 | - |
| FineCLIP | ViT-B/16 | 48.4 | 23.3 | - |
| SigLIP 2 | ViT-B/16 | 53.4 | 20.6 | 57.9 |
| FG-CLIP | ViT-B/16 | 52.3 | 28.6 | - |
| FG-CLIP 2 | ViT-B/16 | **74.9** | **47.3** | **60.7** |
| CLIP | ViT-L/14 | 33.8 | 9.3 | - |
| EVA-CLIP | ViT-L/14 | 32.1 | 18.3 | - |
| Long-CLIP | ViT-L/14 | 35.6 | 10.4 | - |
| FineCLIP | ViT-L/14 | 54.5 | 22.5 | - |
| SigLIP 2 | ViT-L/16 | 54.7 | 25.9 | 56.6 |
| CLOC | ViT-L/14 | 72.9 | 32.6 | - |
| FG-CLIP | ViT-L/14 | 63.2 | 38.3 | - |
| FG-CLIP 2 | ViT-L/16 | **74.0** | **41.9** | **68.6** |
| Meta CLIP 2 | ViT-H/14 | 52.0 | 24.4 | 55.2 |
| SigLIP 2 | ViT-So/16 | 62.0 | 31.4 | 63.6 |
| FG-CLIP 2 | ViT-So/16 | **77.4** | **43.9** | **66.5** |

Each region is paired with one positive description and ten synthetically perturbed negatives, forming a challenging discrimination task. The benchmark comprises four subsets: trivial, easy, medium, and hard, arranged by increasing linguistic subtlety between correct and distractor texts, requiring finer-grained reasoning for accurate matching. Following FineCLIP (Jing et al., 2024), we extract dense visual features and use ROIAlign with provided bounding boxes to obtain region-specific representations. Similarity scores are computed between region features and textual descriptions, with Top-1 accuracy used as the evaluation metric. Results in Table 1 show that FG-CLIP 2 achieves significant gains over prior models. This demonstrates its superior capability in distinguishing subtle visual-linguistic correspondences, a key requirement for fine-grained understanding.

*Table 3.* Open-vocabulary object detection results on LVIS$^{\text{minival}}$ and LVIS. AP is reported across all categories and frequency splits.

| Method | Backbone | LVIS$^{\text{minival}}$ | | | | LVIS | | | |
|---|---|---|---|---|---|---|---|---|---|
| | | AP | AP$_r$ | AP$_c$ | AP$_f$ | AP | AP$_r$ | AP$_c$ | AP$_f$ |
| YOLO-World-L (Cheng et al., 2024) | YOLOv8-L | 35.5 | 24.4 | 34.0 | 38.8 | 28.7 | 22.9 | 24.9 | 35.4 |
| OWL-ST (Ye et al., 2023) | ViT-B/16 | 34.4 | 38.3 | – | – | 28.6 | 30.3 | – | – |
| DetCLIP v3 (Yao et al., 2024) | Swin-T | 47.0 | 45.1 | 47.7 | 46.7 | 38.9 | 37.2 | 37.5 | 41.2 |
| T-Rex2 (Jiang et al., 2024) | Swin-T | 42.8 | 37.4 | 39.7 | 46.5 | 34.8 | 29.0 | 31.5 | 41.2 |
| OV-DINO (Wang et al., 2024) | Swin-T | 40.1 | 34.5 | 39.5 | 44.1 | 32.9 | 29.1 | 30.4 | 37.4 |
| LLMDet | Swin-T | 44.7 | 37.3 | 39.5 | 50.7 | 34.9 | 26.0 | 30.1 | 44.3 |
| GLIP (Li et al., 2022) | Swin-L | 37.3 | 28.2 | 34.3 | 41.5 | 26.9 | 17.1 | 23.3 | 36.4 |
| Grounding-DINO (Liu et al., 2023b) | Swin-L | 33.9 | 22.2 | 30.7 | 38.8 | – | – | – | – |
| OWL-ST (Ye et al., 2023) | ViT-L/14 | 40.9 | 41.5 | – | – | 35.2 | 36.2 | – | – |
| MM-GDINO (Zhao et al., 2024) | Swin-L | 36.8 | 28.1 | 31.8 | 42.8 | 29.1 | 19.7 | 25.6 | 37.2 |
| LLMDet | Swin-L | 51.1 | 45.1 | 46.1 | 56.6 | 42.0 | 31.6 | 38.8 | 50.2 |
| LLMDet + FG-CLIP | Swin-T + ViT-B/16 | 48.0 | 40.6 | 47.7 | 51.4 | 41.0 | 35.1 | 38.9 | 45.8 |
| LLMDet + SigLIP 2 | Swin-T + ViT-B/16 | 47.9 | 42.4 | 45.6 | 51.0 | 41.8 | 36.1 | 39.9 | 45.2 |
| LLMDet + FG-CLIP 2 | Swin-T + ViT-B/16 | **51.6** | **47.5** | **50.7** | **53.2** | **44.0** | **37.4** | **42.8** | **48.2** |
| LLMDet + FG-CLIP | Swin-T + ViT-L/14 | 50.5 | 41.9 | 49.3 | 53.1 | 43.1 | 37.0 | 41.3 | 47.7 |
| LLMDet + SigLIP 2 | Swin-T + ViT-L/16 | 49.9 | 46.9 | 48.9 | 51.3 | 43.6 | 40.4 | 42.0 | 45.9 |
| LLMDet + FG-CLIP 2 | Swin-T + ViT-L/16 | **52.6** | **48.6** | **51.8** | **54.0** | **45.5** | **41.0** | **44.2** | **49.0** |
| LLMDet + SigLIP 2 | Swin-T + ViT-So/16 | 50.2 | 48.4 | 49.0 | 51.7 | 44.3 | 41.1 | 42.7 | 46.4 |
| LLMDet + Meta CLIP 2 | Swin-T + ViT-H/14 | 52.2 | 50.5 | 51.1 | 53.5 | 44.5 | 41.7 | 42.9 | 47.6 |
| LLMDet + FG-CLIP 2 | Swin-T + ViT-So/16 | **53.1** | **50.8** | **52.3** | **54.2** | **45.9** | **42.1** | **44.6** | **49.0** |

*Table 4.* Comparisons on English image-level tasks, including long/short caption image-text retrieval, and zero-shot image classification.

| Method | Backbone | ShareGPT4V | | DCI | | MSCOCO | | Flickr30k | | IN-1K | IN-v2 |
|---|---|---|---|---|---|---|---|---|---|---|---|
| | | I→T | T→I | I→T | T→I | I→T | T→I | I→T | T→I | Top-1 | Top-1 |
| CLIP | ViT-B/16 | 78.2 | 79.6 | 45.5 | 43.0 | 51.8 | 32.7 | 82.2 | 62.1 | 68.4 | 61.9 |
| EVA-CLIP | ViT-B/16 | 90.5 | 85.5 | 41.9 | 41.2 | 58.7 | 41.6 | 85.7 | 71.2 | 74.7 | 67.0 |
| Long-CLIP | ViT-B/16 | 94.7 | 93.4 | 51.7 | 57.3 | 57.6 | 40.4 | 85.9 | 70.7 | 66.8 | 61.2 |
| FineCLIP | ViT-B/16 | 70.6 | 73.3 | 35.5 | 34.4 | 54.5 | 40.2 | 82.5 | 67.9 | 55.7 | 48.8 |
| UMG-CLIP | ViT-B/16 | - | - | - | - | 64.7 | 51.6 | 91.4 | 78.6 | - | 66.5 |
| FG-CLIP | ViT-B/16 | **96.7** | 94.9 | 61.8 | 60.6 | 64.1 | 45.4 | 90.7 | 76.4 | 69.0 | 61.8 |
| SigLIP 2 | ViT-B/16 | 66.0 | 67.9 | 32.3 | 34.2 | 71.2 | **55.2** | 92.6 | 78.0 | **81.2** | **74.5** |
| FG-CLIP 2 | ViT-B/16 | 95.8 | **95.4** | **64.5** | **64.9** | **72.1** | 54.5 | **94.1** | **81.9** | 79.5 | 72.2 |
| CLIP | ViT-L/14 | 86.5 | 83.6 | 37.2 | 36.4 | 58.0 | 37.1 | 87.4 | 67.3 | 76.6 | 70.9 |
| EVA-CLIP | ViT-L/14 | 91.5 | 89.4 | 47.2 | 47.8 | 64.2 | 47.9 | 89.2 | 77.9 | 80.4 | 73.8 |
| Long-CLIP | ViT-L/14 | 95.8 | 95.6 | 44.2 | 52.5 | 62.8 | 46.3 | 90.0 | 76.2 | 73.5 | 67.9 |
| FineCLIP | ViT-L/14 | 73.4 | 82.7 | 40.1 | 46.2 | - | - | - | - | 60.8 | 53.4 |
| UMG-CLIP | ViT-L/14 | - | - | - | - | 68.9 | 54.6 | 93.4 | 83.1 | - | 71.6 |
| FG-CLIP | ViT-L/14 | **97.4** | **96.8** | 66.7 | 66.1 | 68.9 | 50.9 | 93.7 | 81.5 | 76.1 | 69.0 |
| CLOC | ViT-L/14 | - | - | - | - | 74.8 | 54.4 | - | - | 80.1 | 73.2 |
| SigLIP 2 | ViT-L/16 | 85.1 | 84.6 | 48.0 | 49.3 | 72.1 | 55.2 | 94.3 | 82.6 | **83.1** | 76.5 |
| FG-CLIP 2 | ViT-L/16 | 96.9 | 96.6 | **70.0** | **71.6** | **75.1** | **58.6** | **96.6** | **84.8** | 83.0 | **77.4** |
| SigLIP 2 | ViT-So/16 | 78.6 | 79.5 | 46.0 | 47.1 | 71.0 | 55.8 | 94.1 | 82.5 | 83.8 | 77.7 |
| Meta CLIP 2 | ViT-H/14 | 93.9 | 89.2 | 53.0 | 50.2 | 66.8 | 47.7 | 91.9 | 77.0 | 81.7 | 75.7 |
| FG-CLIP 2 | ViT-So/16 | **97.5** | **96.7** | **70.6** | **72.1** | **74.6** | **56.7** | **95.9** | **85.0** | **84.1** | **77.8** |

### 4.2.2. BOUNDING BOX CLASSIFICATION

We evaluate zero-shot bounding box classification on COCO-val2017 (Lin et al., 2014), LVIS (Gupta et al., 2019), and our proposed BoxClass-CN dataset, following the protocol of (Xie et al., 2025). While COCO and LVIS focus on English category recognition within localized regions, BoxClass-CN targets Chinese, enabling a bilingual assessment of fine-grained vision-language alignment. As shown in Table 2, FG-CLIP 2 achieves state-of-the-art performance in both languages and significantly outperforms all compared open-source models. These results demonstrate its robust ability to align local visual content with semantic concepts across linguistic boundaries.

### 4.2.3. OPEN-VOCABULARY OBJECT DETECTION

To assess the impact of vision-language alignment models on open-vocabulary detection (OVD), we adopt a training-free evaluation protocol that avoids biases from detector fine-tuning. Unlike prior approaches (Wu et al., 2024) that require small-scale detector training on fixed categories, our method directly leverages the zero-shot generalization of alignment models to calibrate confidence scores and cate-

*Table 5.* Performance on Chinese image-text retrieval benchmarks, covering both long-text and short-text settings. Results are reported in terms of Recall@1 (%).

| Method | Backbone | LIT-CN | | DCI-CN | | DOCCI-CN | | Flickr-CNA | | COCO-CN | |
|---|---|---|---|---|---|---|---|---|---|---|---|
| | | I→T | T→I | I→T | T→I | I→T | T→I | I→T | T→I | I→T | T→I |
| R2D2 | ViT-B/16 | 35.7 | 27.4 | 25.9 | 27.3 | 36.1 | 36.9 | 69.7 | 51.1 | 60.1 | 45.5 |
| Chinese-CLIP | ViT-B/16 | 45.7 | 35.6 | 30.1 | 27.9 | 44.6 | 43.1 | 75.8 | 62.4 | 68.8 | 54.9 |
| SigLIP 2 | ViT-B/16 | 4.6 | 2.6 | 5.0 | 4.0 | 7.6 | 5.7 | 71.7 | 49.1 | 68.6 | 46.2 |
| FG-CLIP 2 | ViT-B/16 | **82.4** | **81.1** | **53.9** | **55.7** | **71.2** | **75.4** | **85.4** | **69.9** | **77.2** | **62.9** |
| R2D2 | ViT-L/14 | 48.3 | 33.3 | 35.6 | 34.2 | 49.5 | 46.3 | 78.8 | 60.0 | 69.6 | 52.7 |
| Chinese-CLIP | ViT-L/14 | 48.6 | 38.9 | 31.4 | 32.7 | 49.7 | 50.8 | 82.9 | 69.6 | 74.3 | 59.9 |
| SigLIP 2 | ViT-L/16 | 14.8 | 10.9 | 13.6 | 14.4 | 24.6 | 27.3 | 79.8 | 53.2 | 74.2 | 51.7 |
| FG-CLIP 2 | ViT-L/16 | **86.3** | **85.9** | **60.4** | **62.2** | **77.6** | **81.9** | **90.3** | **75.0** | **82.8** | **66.5** |
| SigLIP 2 | ViT-So/16 | 17.0 | 10.8 | 13.4 | 12.0 | 25.0 | 21.3 | 78.4 | 51.7 | 72.0 | 50.7 |
| Meta CLIP 2 | ViT-H/14 | 77.2 | 67.6 | 53.8 | 52.1 | 73.8 | 77.2 | 89.3 | 72.2 | 80.1 | 63.1 |
| FG-CLIP 2 | ViT-So/16 | **87.6** | **86.3** | **62.7** | **65.1** | **79.7** | **84.0** | **91.5** | **77.2** | **83.2** | **68.1** |

gory predictions in the final OVD output. This allows for a cleaner analysis of the contribution of image-text alignment to OVD performance. We adopt a fusion strategy that leverages the vision-language alignment model to refine the class predictions and confidence scores of a pre-trained detector. Specifically, similarity scores from the alignment model are combined with the detector's original confidences via geometric averaging, enabling more semantically accurate and calibrated open-vocabulary detection. We use LLMDet (Fu et al., 2025) as the base detector and evaluate on the challenging LVIS dataset (Gupta et al., 2019), which contains 1,203 categories. As shown in Table 3, FG-CLIP 2 combined with LLMDet achieves the best performance among open-source methods, demonstrating its strong practical utility and superior generalization in detection scenarios.

### 4.3. Image-level tasks

#### 4.3.1. LONG/SHORT CAPTION IMAGE-TEXT RETRIEVAL

To comprehensively evaluate image-text alignment under varying linguistic complexity, we conduct experiments on both short and long caption retrieval tasks. Short-text retrieval (Flickr30k (Young et al., 2014), MSCOCO (Lin et al., 2014)) assesses basic semantic matching, while long-text retrieval requires fine-grained understanding of detailed descriptions and complex visual scenes. For short-text retrieval, we employ the validation set of MSCOCO and the test set of Flickr30k, which are widely used benchmarks for assessing image-text alignment models. For long-text retrieval in English, we use the 1K subset of ShareGPT4V (Chen et al., 2024b) and the full test set of DCI (Urbanek et al., 2024) following the protocol of Long-CLIP (Zhang et al., 2024a). For Chinese, we evaluate on LIT-CN, DCI-CN, and DOCCI-CN for long-text, and Flickr-CNA (Xie et al., 2023) and COCO-CN (Li et al., 2019) for short-text. We employ the validation set of COCO-CN and the test set of Flickr-CNA. These datasets cover diverse content and caption styles, providing a robust evaluation

*Table 6.* Performance on dense prediction tasks.

| Model | Backbone | A-847 | PC-459 | A-150 | PC-59 | VOC-20 | VOC-21 |
|---|---|---|---|---|---|---|---|
| CLIP | ViT-B/16 | 8.4 | 16.6 | 27.2 | 57.5 | 93.7 | 78.3 |
| CLIPSelf | ViT-B/16 | 10.1 | - | 29.7 | 55.3 | - | - |
| FineCLIP | ViT-B/16 | 12.2 | - | 32.4 | 56.0 | - | - |
| UMG-CLIP | ViT-B/16 | 13.8 | 21.1 | 34.6 | 58.2 | - | - |
| FG-CLIP | ViT-B/16 | 12.3 | 19.1 | 33.4 | 58.2 | 95.3 | 77.9 |
| SigLIP 2 | ViT-B/16 | 10.4 | 17.0 | 28.5 | 55.4 | 94.4 | 75.8 |
| FG-CLIP 2 | ViT-B/16 | **16.6** | **24.0** | **38.5** | **61.2** | **97.1** | **81.1** |
| CLIP | ViT-L/14 | 10.8 | 20.4 | 31.5 | 62.0 | 96.6 | 81.8 |
| CLIPSelf | ViT-L/14 | 13.6 | - | 34.9 | 59.1 | - | - |
| FineCLIP | ViT-L/14 | 14.1 | - | 36.1 | 59.9 | - | - |
| UMG-CLIP | ViT-L/14 | 15.4 | 23.2 | 36.1 | 58.7 | - | - |
| FG-CLIP | ViT-L/14 | 14.6 | 23.3 | 36.9 | 61.4 | 97.4 | 81.8 |
| SigLIP 2 | ViT-L/16 | 14.3 | 24.1 | 38.8 | 62.4 | 97.0 | 82.3 |
| FG-CLIP 2 | ViT-L/16 | 18.8 | 26.6 | 41.2 | 62.4 | 97.6 | 81.8 |
| FG-CLIP 2 | ViT-So/16 | **20.0** | **27.5** | **42.2** | **63.3** | **97.8** | **83.2** |

of multilingual performance. As shown in Table 4 and Table 5, FG-CLIP 2 achieves consistent improvements across all settings, with particularly strong gains on long-text retrieval, highlighting its superior capability in fine-grained vision-language alignment. Notably, FG-CLIP 2 outperforms Meta CLIP 2 (Chuang et al., 2025), the current multilingual SOTA, on both language settings, despite using a smaller ViT-L/16 backbone with 1.0 billion parameters compared to Meta CLIP 2's 1.8 billion parameter ViT-H/14. This highlights the effectiveness of our training paradigm in achieving stronger performance with reduced model scale.

#### 4.3.2. ZERO-SHOT IMAGE CLASSIFICATION

We evaluate zero-shot image classification on ImageNet-1K (Deng et al., 2009) and ImageNet-v2 (Recht et al., 2019) using standard prompts (Radford et al., 2021). As shown in Table 4, FG-CLIP 2 achieves competitive performance compared to SigLIP 2 (Tschannen et al., 2025), and outperforms EVA-CLIP (Sun et al., 2023), Long-CLIP (Zhang et al., 2024a), FineCLIP (Jing et al., 2024), UMG-CLIP (Shi et al., 2024), CLOC (Chen et al., 2024a), and Meta CLIP 2. This confirms that the improvements in fine-grained alignment do not come at the cost of standard recognition accuracy,

*Table 7.* Comparisons on large multimodal model benchmarks.

| Method | GQA | MMMU | TextVQA | RefCOCO | | | MMBench-EN | | MMBench-CN | | V* Benchmark | | |
|---|---|---|---|---|---|---|---|---|---|---|---|---|---|
| | | | | Val | TestA | TestB | Dev | Test | Dev | Test | AR | SRR | Overall |
| LLaVA-1.5 + CLIP | 61.9 | 35.7 | 58.2 | 76.2 | 83.4 | 67.9 | 65.1 | 66.5 | 58.2 | 58.4 | 44.4 | 52.6 | 47.6 |
| LLaVA-1.5 + SigLIP 2 | 62.0 | 37.0 | 56.6 | 79.5 | 84.4 | 73.9 | 66.2 | 64.8 | 58.0 | 57.2 | 40.9 | 56.6 | 47.1 |
| LLaVA-1.5 + Meta CLIP 2 | 62.4 | 37.0 | 59.4 | 76.7 | 82.8 | 69.8 | 66.5 | **67.0** | 59.3 | 60.3 | 39.1 | 54.0 | 45.0 |
| LLaVA-1.5 + FG-CLIP 2 | **64.0** | **38.1** | **62.0** | **84.9** | **89.8** | **79.5** | **67.6** | 66.5 | **60.5** | **61.4** | **45.2** | **57.9** | **50.3** |

*Table 8.* Ablation study results for the training objectives of FG-CLIP 2.

| Method | COCO$^{80}$ | | | FG-OVD | | | Flickr30k | | ImageNet-V2 |
|---|---|---|---|---|---|---|---|---|---|
| | Top-1 | Top-5 | Hard | Medium | Easy | Trivial | I→T | T→I | Top-1 |
| FG-CLIP 2 | 74.9 | 95.7 | 52.3 | 76.3 | 80.3 | 92.0 | 94.1 | 81.9 | 72.2 |
| w/o $\mathcal{L}_{\text{CMR}}$ | 74.0 | 94.9 | 50.9 | 75.1 | 77.8 | 93.5 | 93.3 | 81.9 | 72.1 |
| w/o $\mathcal{L}_{\text{TIC}}$ | 70.1 | 94.8 | 51.6 | 75.1 | 79.1 | 92.1 | 93.7 | 81.8 | 72.1 |
| w/o $\mathcal{L}_{\text{CMR}}$, $\mathcal{L}_{\text{TIC}}$ | 62.7 | 93.4 | 51.7 | 76.0 | 77.6 | 90.7 | 93.5 | 81.6 | 72.0 |

demonstrating a well-balanced representation capability.

### 4.4. Dense prediction tasks

We evaluate dense prediction through open-vocabulary segmentation, a task that requires models to segment object categories beyond a fixed training set. We adopt Cat-Seg (Cho et al., 2024) as the base framework, trained on COCO-Stuff-164k (172 categories), and evaluate on datasets with diverse category schemas: ADE20k (847 or 150 categories, denoted A-847/A-150), Pascal Context (PC-459/PC-59), and Pascal VOC (VOC-20/VOC-21). As shown in Table 6, FG-CLIP 2 achieves the best performance across models of various scales, delivering consistent improvements over the baseline. This demonstrates its strong capability in enabling pixel-level generalization, crucial for downstream dense understanding tasks.

### 4.5. Large Multimodal Model tasks

We empoly FG-CLIP 2 as a vision encoder in large multimodal models (LMMs) to assess its compatibility and utility in advanced multimodal reasoning. We integrate FG-CLIP 2 into a standard LLaVA-style LMM architecture, following the pre-training and supervised fine-tuning protocol of LLaVA-1.5 (Liu et al., 2023a). Evaluation is conducted on a diverse set of benchmarks spanning visual question answering (GQA (Hudson & Manning, 2019), TextVQA (Singh et al., 2019)), multimodal understanding (MMMU (Yue et al., 2024)), general multimodal evaluation (MMBench-EN and MMBench-CN (Liu et al., 2024)), referring expression comprehension (RefCOCO (Kazemzadeh et al., 2014)), and fine-grained visual reasoning (V* Benchmark (Wu & Xie, 2024)). Results in Table 7 show that LMMs equipped with FG-CLIP 2 outperform those using other open-source encoders, including SigLIP 2 and Meta CLIP 2. This indicates that the fine-grained and bilingual capabilities of

FG-CLIP 2 effectively transfer to higher-level multimodal tasks, making it a strong candidate for integration into next-generation LMMs.

### 4.6. Ablation Study

As our training framework leverages global-level, region-level, and hard-negative data through complementary learning mechanisms, we adopt a baseline model that incorporates global alignment alongside fine-grained visual and textual learning. This allows us to specifically evaluate the impact of the Cross-modal Rank Loss with Global Threshold Synchronization ($\mathcal{L}_{\text{CMR}}$) and the Textual Intra-modal Contrastive Loss ($\mathcal{L}_{\text{TIC}}$). As shown in Table 8, removing $\mathcal{L}_{\text{TIC}}$ leads to a 4.8-point drop in COCO Top-1 accuracy (to 70.1%) and a decrease in FG-OVD Hard performance from 52.3% to 51.6%, confirming its critical role in distinguishing semantically similar texts. Removing $\mathcal{L}_{\text{CMR}}$ also degrades performance, with FG-OVD Hard falling to 50.9%, indicating its importance in cross-modal alignment. When both losses are removed, COCO Top-1 drops sharply to 62.7%, demonstrating their complementary benefits. The full model achieves consistent gains across all tasks, especially in bounding box classification and fine-grained understanding, while maintaining strong results on standard recognition benchmarks, validating the effectiveness of our proposed training objectives. Additionally, visualizations demonstrating the semantic separability improvement from TIC loss are provided in Appendix H.

To examine how different data compositions affect the model's bilingual capability, we conduct an ablation study comparing two variants trained under identical settings except for the data used in the second stage: one uses English-only data, while the other uses both English and Chinese data. Results in Table C (Appendix D) show that the bilingual variant not only achieves higher accuracy on Chinese

benchmarks but also consistently improves performance on English-only evaluation sets, confirming a mutually promoting effect between the two languages. We further conduct a comparison with re-trained baselines under identical training data in Appendix E, showing that FG-CLIP 2's architecture and learning objectives contribute significantly to its performance gains.

## 5. Conclusion and Limitations

In this work, we present FG-CLIP 2, a bilingual vision-language model that advances fine-grained understanding for both English and Chinese. Our two-stage training paradigm progressively refines alignment by leveraging both short and long captions, region-text supervision, and multiple discriminative objectives, including the proposed Textual Intra-modal Contrastive (TIC) loss to better distinguish semantically similar descriptions. Trained on large-scale, high-quality English and Chinese datasets, FG-CLIP 2 achieves superior performance across 29 datasets and 8 tasks, demonstrating strong bilingual generalization. To advance evaluation in non-English settings, we introduce a new benchmark for Chinese multimodal understanding with challenging tasks such as long caption image-text retrieval and bounding box classification. We release the model, code, and benchmark to support future research in bilingual fine-grained vision-language understanding. In future work, we will extend the model to handle longer textual inputs and explicitly model relational structures among objects.

## Acknowledgement

This work was supported by the National Natural Science Foundation of China under Grants No. 62076019.

## Impact Statement

This paper aims to advance the field of Machine Learning, which has broad implications for society. There are many potential societal consequences of our work, none which we feel must be specifically highlighted here.

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

## A. Training Data Details

Table A summarizes the datasets used in our two-stage training pipeline. We present examples of the Chinese and English training data used in Stage II in Figure A.

*Table A.* Overview of the training datasets used in two stages.

| Data Type | Dataset | Language | Size |
|---|---|---|---|
| Image-text | LAION-2B-enhanced | English | 1.6B |
| | Wukong | Chinese | 100M |
| | Zero | Chinese | 250M |
| | In-house Data | Chinese | 500M |
| Region-text | FineHARD | English | 12M |
| | FineRegion-CN | Chinese | 12M |

*Table B.* Detailed parameters for training with Cat-Seg.

| Parameter name | Value | Parameter name | Value |
|---|---|---|---|
| Text Guidance Proj Dim | 128 | Min Size Train | 384 |
| Appearance Guidance Proj Dim | 128 | Min Size Train Sampling | Choice |
| Decoder Dims | [64, 32] | Min Size Test | 640 |
| Decoder Guidance Dims | [256, 128] | Size Divisibility | 384 |
| Decoder Guidance Proj Dims | [32, 16] | Format | RGB |
| Num Layers | 2 | Dataset Mapper Name | Mask Former Semantic |
| Num Heads | 4 | Images Per Batch | 8 |
| Hidden Dims | 128 | LR Scheduler Name | Warmup Cosine LR |
| Pooling Sizes | [2, 2] | Base Learning Rate | 0.0002 |
| Feature Resolution | [24, 24] | Max Iterations | 80,000 |
| Window Sizes | 12 | Backbone Multiplier | 0.0 |
| Attention Type | Linear | CLIP Multiplier | 0.01 |

## B. Open-Vocabulary Object Detection Experimental Details

In our experiments, we adopt LLMDet as the base open-vocabulary object detection model. Its output consists of bounding boxes, each associated with a predicted category and a confidence score. To improve category accuracy, we recalibrate these predictions using vision-language alignment models, such as FG-CLIP 2, without modifying LLMDet's parameters.

For each detected bounding box, we extract its visual feature by applying RoI-Align on the dense ViT feature map produced by FG-CLIP 2. We also encode all candidate category names into text embeddings using the same model. We then compute the cosine similarity between the region's visual feature and each category's text embedding, followed by Softmax normalization to obtain a category-wise alignment similarity distribution. To produce the final prediction, we combine LLMDet's original confidence score with FG-CLIP 2's normalized similarity score via geometric averaging. The resulting fused score reflects both the detector's localization confidence and the alignment model's semantic relevance. We then select the category with the highest fused score as the final predicted class, and assign the corresponding fused value as the final confidence output.

This approach leverages FG-CLIP 2's fine-grained understanding to recalibrate LLMDet's predictions across the entire category space. It ensures that categories with strong semantic alignment but initially low detection scores can still be correctly selected if their fused confidence is highest. This global recalibration significantly improves performance on novel categories, while maintaining compatibility with the original detector's structure.

## C. Dense Prediction Tasks Experimental Details

We adopt Cat-Seg (Cho et al., 2024) as the base model for open-vocabulary segmentation, which supports plug-and-play integration of various image-text models. A unified training configuration is used across different ViT backbones, with

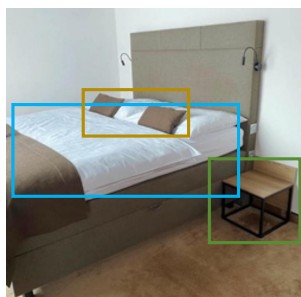

---------long caption---------
The image presents a bedroom scene centered on a large bed with a tall, upholstered headboard. The headboard is light brown and has a smooth texture without any visible patterns. Two reading lights are mounted on the wall on each side of the headboard, extending outward with a flexible arm and a small light at the end. These lights are positioned at a height that seems suitable for reading. The bed frame is also light brown and has a contemporary design.. . On the bed, there is a white duvet and a couple of pillows, with one of them being a smaller pillow that matches the color of the reading lights, and the other being a larger pillow that appears to have a different color, possibly a neutral tone. At the foot of the bed, there is a small wooden side table with a black metal frame, which is a minimalistic design with a flat surface and a single visible drawer. The table's color palette includes brown and black, complementing the overall room tones.. . The lighting within the image is not discernible due to the lack of shadows or highlights that could indicate a specific light source. Instead, the room appears to be lit by ambient light that creates a soft glow, giving the space a cozy atmosphere.. . The style of the image is a photograph. The edges and shadows suggest a natural light source and the clarity of the objects points towards a high-resolution image captured by a camera.
---------short caption---------
Modern bedside table in a simple style in the hotel room – metal furniture
---------region caption---------
a small wooden side table with a black metal frame which has a minimalistic design with a flat surface and a single visible drawer
---------region caption---------
a white duvet and a couple of pillows with one of them
---------region caption---------
a smaller pillow that matches the color of the reading lights and a larger that appears to have a different color

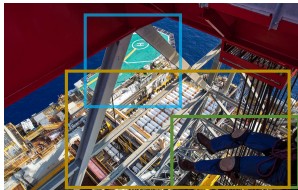

---------long caption---------
This image depicts an aerial view of an oil rig's platform. In the foreground, there is a worker engaged in an activity on the edge of the platform, wearing safety gear such as jeans, a yellow hard hat, knee pads, and brown work boots. The worker appears to be suspended or working at height, secured by a safety harness and other equipment. The rig's infrastructure is comprised of metal beams, pipes, and various work platforms.. . The background reveals a large body of water, which could be the ocean. On the water's surface, there is a green helipad with a circular symbol, suggesting this is a designated area for landing helicopters. The helipad is connected to the rig by a walkway. The weather appears to be clear, with bright sunlight casting shadows on the structure of the rig. The image is taken from a high vantage point, looking downwards towards the worker and the rig's platform, highlighting the scale and complexity of the offshore oil operation.. . The style of the image is a high-resolution photograph, capturing the intricate details of the industrial setting. There are no people or characters other than the worker, so there are no emotions to convey. The description is factual and does not include any subjective interpretations or personal opinions about the image.
---------short caption---------
An overhead view of the legs and harness of an industrial painter suspended from the underside of an offshore rig derrick, high above the deck below.
---------region caption---------
a worker engaged in an activity on the edge of the platform wearing safety gear such as jeans a yellow hard hat knee pads and brown work boots
---------region caption---------
a circular symbol suggesting that this is a designated area for landing helicopters
---------region caption---------
a high vantage point looking downwards towards the worker and the rig's platform highlighting the scale and complexity of the offshore oil operation

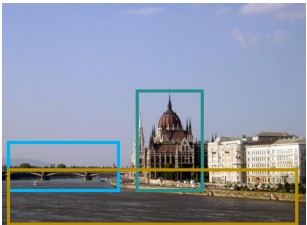

---------long caption---------
这幅图像呈现出一座欧洲风格大型城市的美丽景致，画面的主体是一座哥特复兴风格的宏伟建筑，拥有尖塔、飞扶壁以及巨大的圆顶，坐落在一条平静河流的岸边。从其庄严华丽的建筑风格来看，这座建筑很可能属于议会或政府综合建筑群。
背景天空晴朗，点缀着轻薄的云彩，显示出一个阳光明媚、天气宜人的白天。画面前景中的河面十分平静，可见一艘小船，为画面增添了一丝悠闲与宁静的氛围。远处可以看到一座横跨河流的桥梁，将城市的两岸紧密相连，丰富了整体的城市景观。从光线来看，建筑右侧有柔和的阴影，推测拍摄时间可能在正午附近，阳光较高且明亮。这张图像呈现为高分辨率摄影风格，不仅展现了建筑的复杂细节，也清晰捕捉了天空与水面的柔和质感。画面中没有人物，重点在于建筑之美与自然环境的和谐共存。图像传达的情绪是一种庄严、宏伟与宁静的结合，既体现了历史建筑的力量与工程智慧，又融入了水面与蓝天带来的平静与悠然。
---------short caption---------
布达佩斯：从河上的船只视角拍摄的议会大厦另一景观。
---------region caption---------
一座位于平静河岸的建筑，带有尖塔、飞扶壁和巨大的圆顶。
---------region caption---------
一座远处的桥梁连接着城市的两岸，完善了整个城市景观。
---------region caption---------
平静河流

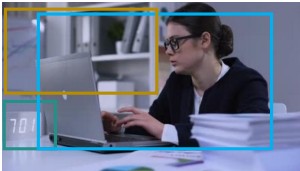

---------long caption---------
这幅图像呈现的是一个专业办公场景。一位穿深色西装外套、戴着眼镜的人正在专注地使用笔记本电脑工作。画面左侧有一个时钟，时间显示为 7:01，暗示可能处于加班时段或面对紧迫的工作截止时间。桌面上摆放着一些文件和文件夹，表明正在处理工作或研究任务。背景中可以看到一个书架，上面摆放着多个文件夹，以及一块白板，上面有一些图示或书写痕迹，进一步烘托出典型办公环境的氛围。整幅图采用真实、高清的摄影风格，光线柔和，细节丰富，对物体的质感和阴影呈现自然。画面中没有其他人物，也没有任何互动动作或文字信息。人物保持端坐姿态，双手放在笔记本电脑的键盘上，表现出专注、投入的工作状态。光线来自柔和的环境光源，可能是天花板灯或办公室台灯，没有明显的强烈阴影。整体氛围体现出一种高效、专注、富有工作压力但又专业有序的办公状态。
---------short caption---------
股票类素材描述：一位女办公室员工因工作量过大感到不开心，没有时间休息。
---------region caption---------
时钟，时间显示为 7:01
---------region caption---------
一位身穿深色西装外套、戴着眼镜的人正在专注地使用笔记本电脑工作
---------region caption---------
一个书架，上面摆放着多个文件夹，以及一块白板，上面有一些图示或书写痕迹

*Figure A.* Examples of Bilingual Training Data.

detailed hyperparameters provided in Table B.

## D. Ablation on Bilingual Data Composition

We present the results of our ablation study on bilingual data composition in Stage II. Both models are initialized from the same Stage I checkpoint trained on the full bilingual (English and Chinese) dataset. In Stage II, we compare two settings: (1) English-only training data, and (2) bilingual English-Chinese training data. As shown in Table C, the bilingual variant not only outperforms the English-only variant on Chinese benchmarks but also consistently improves performance on English-only evaluation sets, confirming a mutually promoting effect between the two languages.

*Table C.* Ablation study on data composition for bilingual capability. FG-CLIP 2* (marked with *) uses English-only data in Stage II, while FG-CLIP 2 uses both English and Chinese data.

| Method | Backbone | DCI
I→T / T→I | MSCOCO
I→T / T→I | DOCCI-CN
I→T / T→I | Flickr30k-CNA
I→T / T→I | COCO$^{80}$
Top-1 / Top-5 | Fine-Grained Understanding
Hard / Medium / Easy / Trivial |
|---|---|---|---|---|---|---|---|
| FG-CLIP 2* | ViT-B/16 | 64.4 / 64.7 | 71.4 / 54.5 | 57.8 / 58.6 | 84.5 / 68.0 | 71.3 / 95.2 | 52.2 / 75.9 / 80.1 / 91.0 |
| FG-CLIP 2 | ViT-B/16 | 64.5 / 64.9 | 72.1 / 54.5 | 71.2 / 75.4 | 85.4 / 69.9 | 74.9 / 95.7 | 52.3 / 76.3 / 80.3 / 92.0 |

## E. Comparison with Re-trained Baselines

To ensure a fair comparison and isolate the impact of model architecture and learning objectives from that of training data, we re-train two representative baselines, SigLIP 2 and Chinese-CLIP, using exactly the same Stage II training data as FG-CLIP 2. Due to computational constraints, all models in this study, including FG-CLIP 2, are trained solely on the Stage II dataset without any Stage I pretraining. This setup guarantees identical data conditions across all methods, enabling a fully controlled evaluation focused solely on architectural and objective-level differences.

Table D summarizes the results under this setting. Remarkably, even in the absence of large-scale pretraining, FG-CLIP 2 consistently outperforms both SigLIP 2 and Chinese-CLIP across all benchmarks, including DCI, MSCOCO, DOCCI-CN, Flickr30k-CNA, COCO$^{80}$, and Fine-Grained Understanding. These results indicate that, beyond the benefits of bilingual training data, the architectural design and tailored learning objectives of FG-CLIP 2 play a crucial role in its superior performance. The consistent gains across diverse benchmarks demonstrate that these modeling choices substantially enhance the model's ability to capture fine-grained semantics and align cross-modal representations effectively.

*Table D.* Comparison with retrained baselines using only Stage II data (no Stage I pretraining, indicated by †).

| Method | Backbone | DCI
I→T / T→I | MSCOCO
I→T / T→I | DOCCI-CN
I→T / T→I | Flickr30k-CNA
I→T / T→I | COCO$^{80}$
Top-1 / Top-5 | Fine-Grained Understanding
Hard / Medium / Easy / Trivial |
|---|---|---|---|---|---|---|---|
| Chinese-CLIP† | ViT-B/16 | 26.9 / 25.5 | 47.3 / 31.7 | 48.3 / 47.7 | 80.0 / 65.5 | 41.4 / 69.0 | 16.3 / 33.0 / 37.1 / 73.9 |
| SigLIP 2† | ViT-B/16 | 49.1 / 49.0 | 71.5 / 53.5 | 50.5 / 49.4 | 81.0 / 52.7 | 53.7 / 79.7 | 25.5 / 50.6 / 59.2 / 83.4 |
| FG-CLIP 2† | ViT-B/16 | **60.9 / 62.6** | **72.1 / 53.7** | **65.9 / 69.5** | **84.3 / 66.1** | **71.0 / 95.4** | **51.9 / 76.0 / 80.5 / 90.6** |

## F. Hyperparameter Ablation Study

In addition to the global alignment learning objective, our model incorporates four auxiliary losses: Fine-Grained Visual Learning (FGV), Fine-Grained Textual Learning (FGT), Textual Intra-modal Contrastive Loss (TIC), and Cross-Modal Rank Loss (CMR). The weights for these losses are set using a sequential tuning strategy: starting from the base model trained with only the global alignment loss, we introduce each auxiliary loss one at a time, tune its weight while keeping previously added losses fixed (and excluding any future ones), and select the value that yields the highest average performance on the validation set.

Figure B shows the average performance over different values of each loss hyperparameter, where the average is computed across several representative evaluation tasks: DCI, MSCOCO, DOCCI-CN, Flickr30k-CNA, Bbox Classification (COCO$^{80}$), and Fine-Grained Understanding.

## G. Visualization of Semantic Alignment Capability for Dense Visual Features

We present the alignment capability of FG-CLIP 2 between dense visual features and text in both Chinese and English contexts. The results are shown in Figure C, where warmer colors indicate higher similarity between image regions and the matched text. The precise localization of high-similarity regions across both languages demonstrates FG-CLIP 2's strong bilingual semantic alignment and fine-grained perception capabilities.

## H. Visual Analysis of the Textual Intra-modal Contrastive Loss

We conduct an embedding visualization comparing text representations with and without the TIC loss. As shown in the visualization at Table E.1 and Table E.2, when TIC is enabled, semantically similar but distinct phrases become better separated in the embedding space. This suggests that TIC indeed enhances semantic separability among fine-grained textual

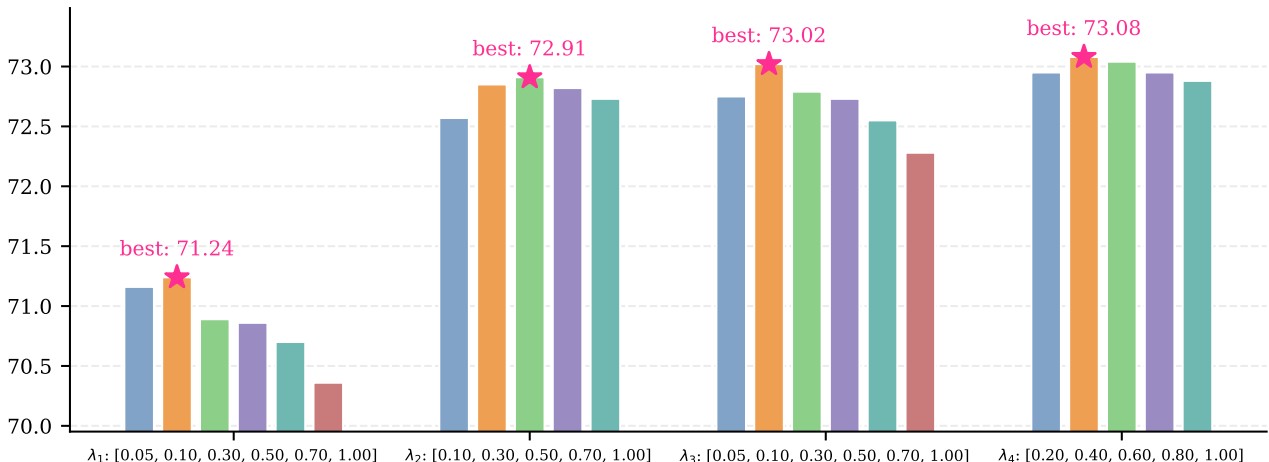

*Figure B.* The performance over different values of each loss hyperparameter.

descriptions.

## I. Examples of LIT-CN

In Table F, we provide examples of long caption image-text pairs from the LIT-CN dataset, covering diverse scene categories such as indoor, outdoor, animals, products, and buildings. These captions not only describe fine-grained subject attributes (e.g., appearance, posture, spatial layout), but also detail the surrounding context, reflecting the dataset's semantic richness and descriptive complexity.

## J. BoxClass-CN Category Schema and Example Explanation

Table G.1 and G.2 provide the complete list of categories in the BoxClass-CN dataset, separated by commas. We further present some examples from BoxClass-CN in Figure D.

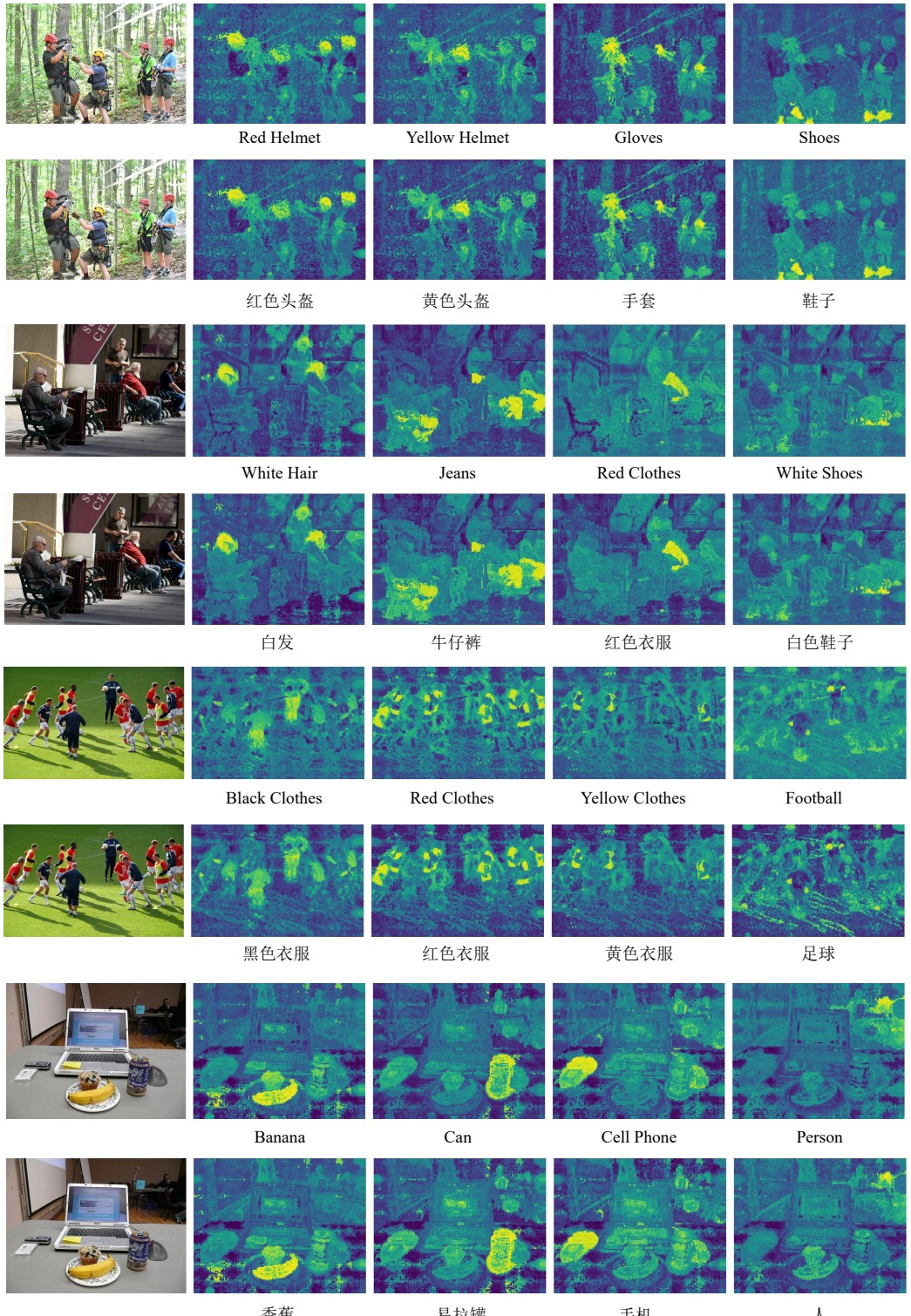

*Figure C.* Visualization of FG-CLIP 2's dense feature maps and semantic alignment capability in bilingual scenarios.

*Table E.1.* The Effectiveness of the Textual Intra-modal Contrastive Loss. (Case1 and Case2)

---

### Case1

T0: "A light grey plastic pipe."  T1: "A light grey fabric pipe."  T2: "A dark pink plastic pipe."
T3: "A light grey text pipe."  T4: "A light pink plastic pipe."  T5: "A green plastic pipe."
T6: "A light grey wood pipe."  T7: "A black plastic pipe."  T8: "A light grey rattan pipe."
T9: "A dark green plastic pipe."  T10: "A purple plastic pipe."

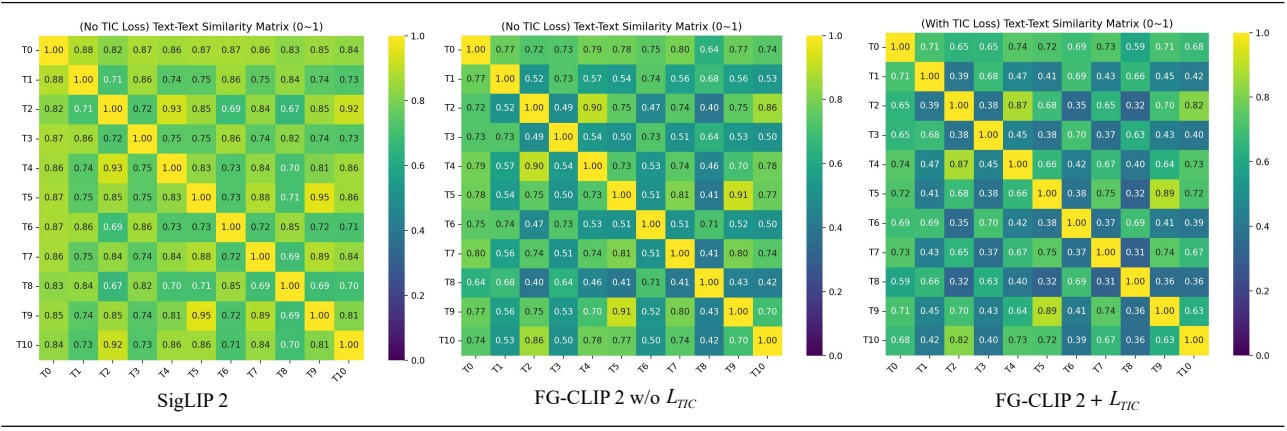

---

### Case2

T0: "一个带有文字图案的浅黄色塑料杯。"  T1: "一个带有点状图案的粉红色塑料杯。"  T2: "一个带有点状图案的蓝色塑料杯。"
T3: "一个浅棕色的图案简单的塑料杯。"  T4: "一个带有格子图案的深棕色塑料杯。"  T5: "一个带有穿孔图案的淡粉色塑料杯。"
T6: "一个带有文字图案的白色玻璃杯。"  T7: "一个印有文字图案的红色纸杯。"  T8: "一个带有条纹图案的深紫色塑料杯。"
T9: "一个带有文字图案的紫色木杯。"  T10: "一个带有铆钉图案的深绿色塑料杯。"

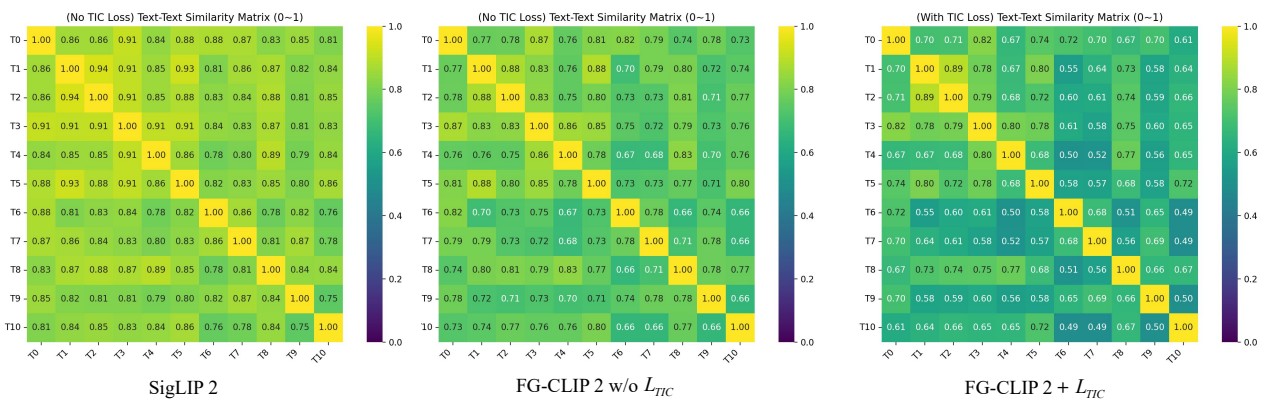

*Table E.2.* The Effectiveness of the Textual Intra-modal Contrastive Loss. (Case3 and Case4)

| Case3 |
| --- |

T0: "一个由布料制成的深红色枕头。"  T1: "一个轻橙色的玻璃枕头。"  T2: "一个暗棕色的纸质枕头。"
T3: "一种由文字组成的浅蓝色枕头。"  T4: "一根轻蓝色的藤制枕头。"  T5: "一个浅灰色的塑料枕头。"
T6: "一个轻紫色的天鹅绒枕头。"  T7: "一个轻紫色的金属枕头。"  T8: "一个蓝色的皮革枕头。"
T9: "一个塑料制成的黄色枕头。"  T10: "一个羊毛制成的棕色枕头。"

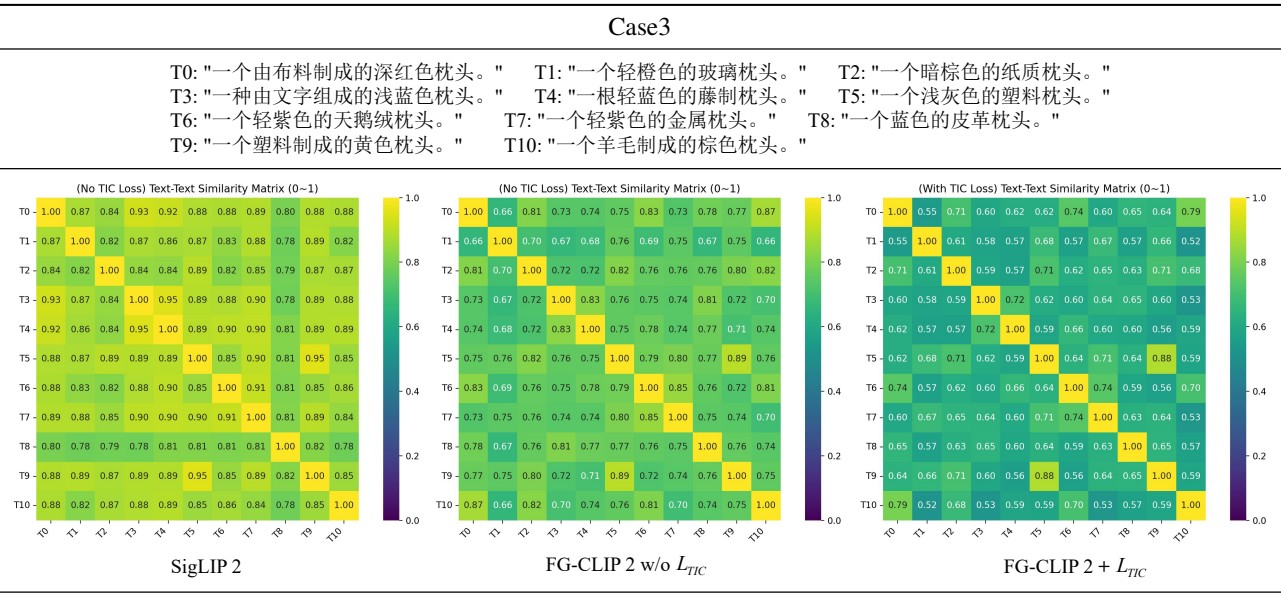

| Case4 |
| --- |

T0: "A black wooden table with a flat surface on top." T1: "A brown rattan table with a flat surface on top." T2: "A dark pink paper table with a flat surface on top."
T3: "A light grey plastic table with a flat surface on top." T4: "A orange leather table with a flat surface on top." T5: "A dark purple velvet table with a flat surface on top."
T6: "A blue stone table with a flat surface on top." T7: "A light pink paper table with a flat surface on top."T8: "A white crochet table with a flat surface on top."
T9: "A purple leather table with a flat surface on top." T10: "A dark red leather table with a flat surface on top."

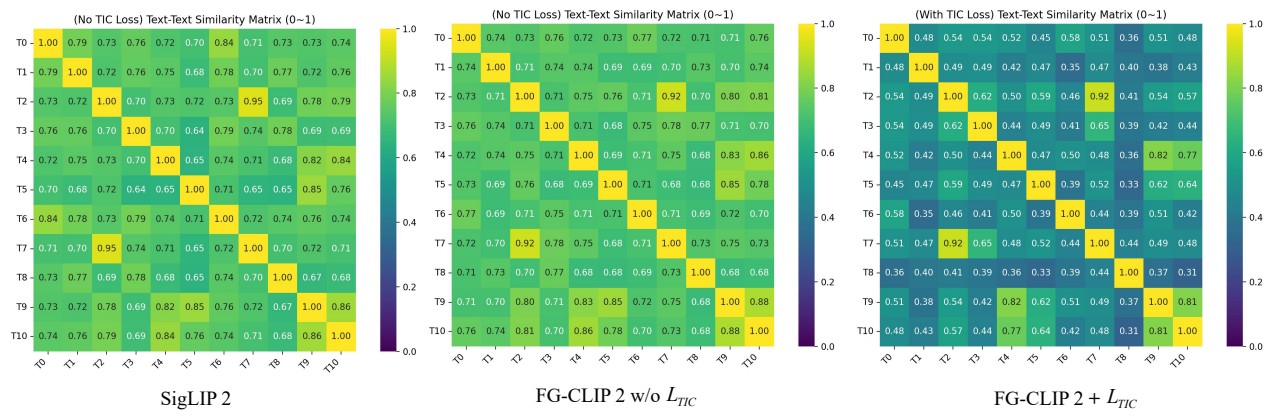

*Table F.* Examples from LIT-CN.

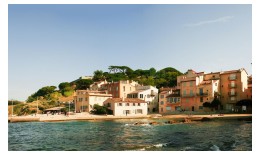

这张图片展示了一片海滨小镇的风景。前景是波光粼粼的海水，水面上有几块岩石。海岸边有几个人在沙滩上活动。背景是一排色彩柔和的建筑，主要为浅黄色、粉色和米色，建筑风格带有地中海风情。建筑后方是绿树覆盖的小山丘，山顶上有一座古老的城堡或堡垒。天空晴朗，呈现出淡蓝色，没有明显的云层。整体画面给人一种宁静、悠闲的感觉。

(This image depicts a seaside town. In the foreground is shimmering water with several rocks scattered across the surface. Along the shore lies a stretch of sandy beach, where a few people are engaged in activities. In the background stands a row of softly colored buildings in shades of light yellow, pink, and beige, featuring Mediterranean-style architecture. Behind the buildings are gently sloping hills covered with green trees, crowned by an ancient castle or fortress at the summit. The sky is clear and pale blue, with no noticeable clouds. The overall scene conveys a sense of tranquility and relaxation.)

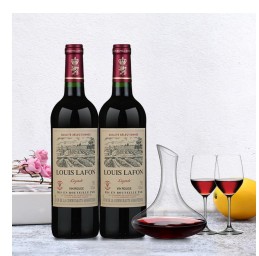

这张图片展示了一组红酒的场景。前景中有两瓶标有"LOUIS LAFON"字样的红酒，瓶身上印有"QUALITÉ SELECTIONNÉE"字样，以及产区和酿酒信息等内容。两瓶红酒均配有深红色的蜡封瓶盖。旁边摆放着一个醒酒器，内有部分红酒，以及两只倒满红酒的透明酒杯。背景中可以看到一个浅灰色的桌面，桌面上还摆放着一小束粉色花卉和一个黄色的柠檬>。整个画面背景为浅色，突出前景物体，风格简洁明了，以实物展示为主，无繁杂装饰。

(This image presents a scene featuring a selection of red wine. In the foreground are two bottles labeled "LOUIS LAFON," with inscriptions including "QUALITÉ SELECTIONNÉE," as well as information about the region and winemaking details. Both bottles are sealed with deep red wax closures. Beside them sits a decanter containing some red wine, along with two clear glasses filled with the wine. In the background, a light gray tabletop is visible, on which a small pink floral arrangement and a yellow lemon are placed. The overall background is light-colored, emphasizing the foreground objects. The style is simple and clean, focusing on the physical presentation of the items with no elaborate decorations.)

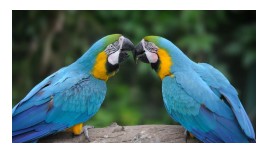

这张图片展示了两只蓝黄金刚鹦鹉。它们的羽毛主要是鲜艳的蓝色，胸部和腹部有黄色的羽毛。头部有绿色的羽毛，眼睛周围有白色和黑色的斑纹。两只鹦鹉面对面，喙部相触，似乎在互动或交流。背景是模糊的绿色，可能是树木或植被。鹦鹉站在一根木头上，姿态自然，显得亲密友好。

(This image shows two blue-and-gold macaws. Their feathers are predominantly bright blue, with yellow plumage on the chest and belly. The head features green feathers, and the area around the eyes has white and black markings. The two birds are facing each other, their beaks touching, appearing to interact or communicate. The background is a blurred green, likely representing trees or foliage. The parrots are perched on a wooden branch, positioned naturally, conveying a sense of closeness and friendliness.)

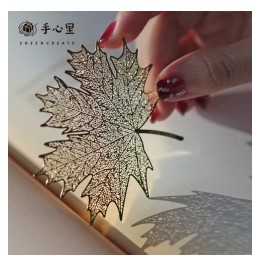

这张图片展示了一只手持金属镂空枫叶书签的特写。书签呈枫叶形状，由细致的金属镂空工艺制成，叶脉清晰，边缘有锯齿状设计。枫叶的主色调为金色，表面反射出一些光线。手指指甲涂有红色指甲油，末端略带光泽，握着书签的柄部。手的背景模糊，焦点集中在手和书签上。背景为一张白色纸张，纸张下方有一条粉色条纹，可能是书封或纸张的一部分。书签的阴影投射在纸张上，形状与书签一致。整体呈现出细腻的手工感和自然的枫叶形态。

(This image features a close-up of a hand holding a metal filigree maple leaf bookmark. The bookmark is shaped like a maple leaf and crafted with intricate metal openwork artistry, showcasing clearly defined veins and serrated edges. It is primarily golden in color, with its surface reflecting subtle highlights. The fingers hold the stem of the bookmark, and the nails are painted red with a slight sheen. The background of the hand is softly blurred, keeping the focus on the hand and the bookmark. The backdrop is a white sheet of paper with a pink stripe along the bottom, likely part of a book cover or decorative paper. The shadow of the bookmark is cast clearly onto the paper, matching its shape. The overall scene conveys fine craftsmanship and the natural beauty of a maple leaf form.)

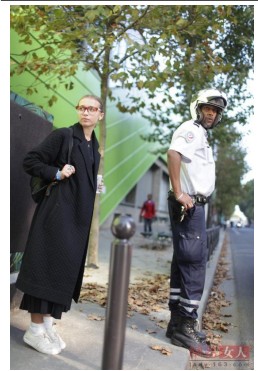

图片展示了一位行人在路边的情景。前景中有两位主要人物：一位是身穿长款黑色风衣、搭配白色运动鞋和红色眼镜的女性，她肩上背着一个黑色包，手中拿着一杯白色的饮料，表情较为认真；另一位是一位穿蓝色和白色制服、戴着白色头盔的交警，他左手拿着一张纸状物品，站立姿态端正，面向前方。背景中有一棵绿叶树木，树叶有一些黄色的枯叶，显示出季节可能是秋季。路边有灰色的金属栏杆，地上散落着一些落叶。远处可以看到绿色的建筑物墙面和一个红色上衣的行人。整体环境为室外的街道路边场景，光线自然。

(The image depicts a scene of pedestrians by the roadside. In the foreground, there are two main figures: one is a woman wearing a long black trench coat, white sneakers, and red glasses. She carries a black bag over her shoulder and holds a white cup containing a beverage, with a serious expression on her face. The other figure is a traffic police officer dressed in a blue and white uniform and wearing a white helmet. He stands upright, facing forward, holding a sheet of paper-like object in his left hand. In the background, there is a tree with green leaves and some yellow, withered foliage, suggesting that the season may be autumn. A gray metal railing runs along the roadside, and fallen leaves are scattered on the ground. In the distance, a green building wall and another pedestrian wearing a red jacket can be seen. The overall setting is an outdoor street-side environment with natural lighting.)

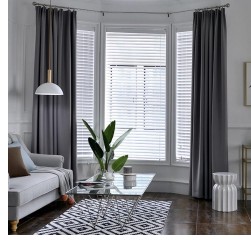

这张图片展示了一个室内的场景，特别是客厅的一部分。前景中有一个浅色沙发，搭配棕色和蓝色抱枕，旁边放置了一条蓝色毯子。沙发旁有一张透明玻璃茶几，茶几上摆放着几本书和一束绿色植物，整体显得简洁而现代。背景是一扇带有白色百叶窗的三角形窗户，窗户两侧是深色窗帘。窗帘上方挂着一根细绳，带有金色环扣，下方悬挂着一盏简约的吊灯。地板上铺有黑白几何图案的地毯，角落里可以看到一个白色装饰凳和部分金色边框的画框。墙面为白色，带有简洁的线条装饰，整体色调以中性色为主，风格简约大气。画面中没有文字或特别的照明效果。

(This image depicts an indoor scene, specifically a part of a living room. In the foreground is a light-colored sofa adorned with brown and blue cushions, accompanied by a blue blanket placed beside it. Adjacent to the sofa stands a transparent glass coffee table, on which several books and a potted green plant are arranged, contributing to a clean and modern aesthetic. In the background is a triangular window fitted with white blinds, flanked by dark curtains on both sides. Above the curtains, a thin cord with golden ring attachments runs across, from which hangs a minimalist pendant lamp. The floor is covered with a black-and-white geometric-patterned rug. In the corner, a white decorative stool and part of a picture frame with a gold trim are visible. The walls are white, accented with simple linear details, and the overall color palette consists mainly of neutral tones, reflecting a sleek and sophisticated style. There is no text or special lighting effects in the image.)

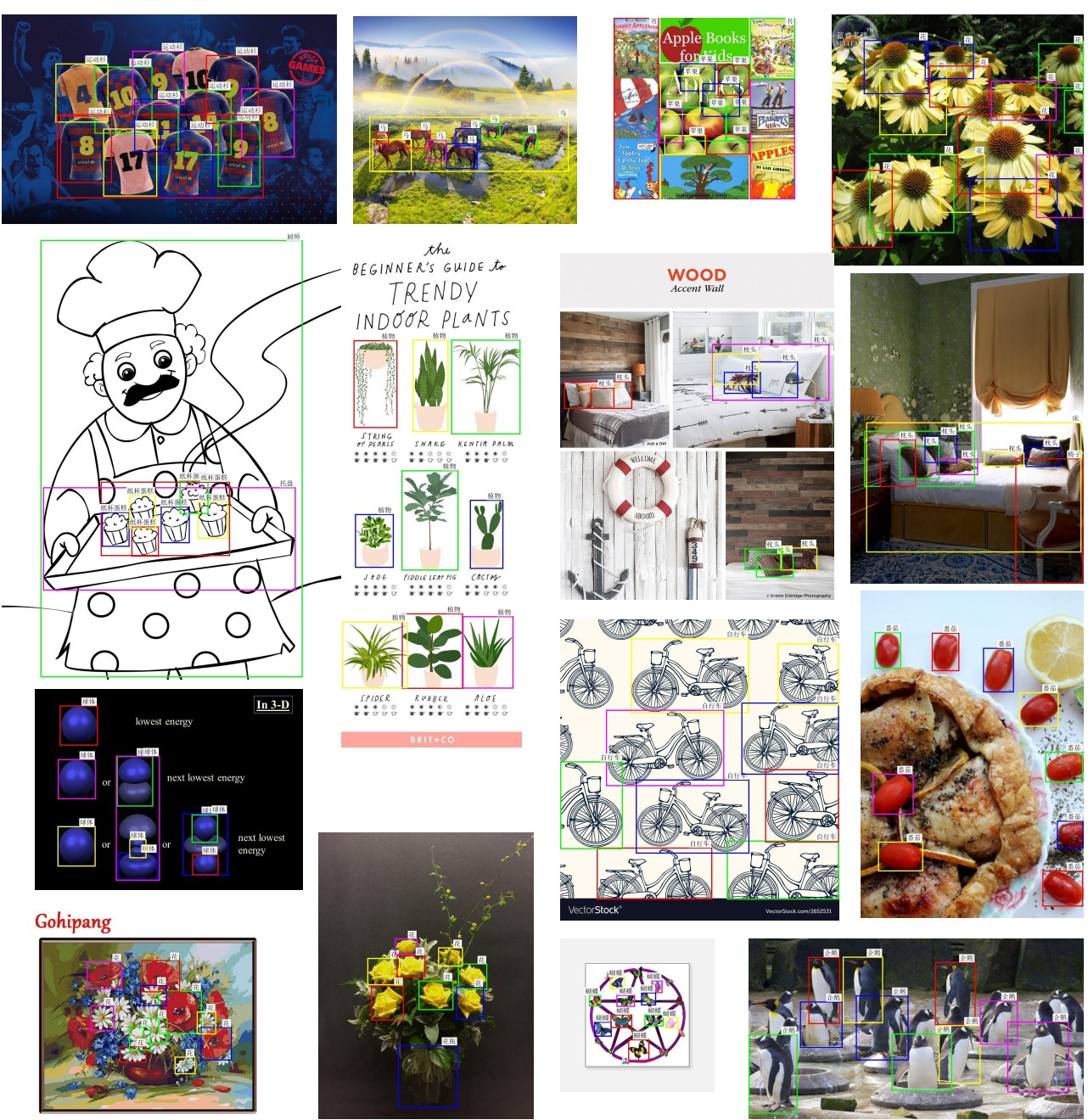

*Figure D.* Examples from BoxClass-CN.

*Table G.1.* All categories in BoxClass-CN, grouped and displayed by ID (1-300).

| id:1-100 |
| --- |
| 连衣裙，书桌，花瓶，花，壁纸，椅子，指甲油，台灯，花瓣，床，衬衫，瓶子，长沙发，微波炉，袋子，眼镜，靴子，自行车，可动人偶，头盔，植物，领结，毯子，胶带，枕头，纸张，领带，吊坠，项链，太阳镜，长裤，房子，屋顶，葡萄酒杯，胡子，耳环，草莓，救生衣，自动售货机，口罩，玩具枪，展台，麦克风，电脑键盘，老鼠，橙子，樱桃，花束，骰子，咖啡，饰物，企鹅，南瓜，护目镜，熊，星星，头发，制服，洗衣机，运动衫，棚屋，院子，横幅，甜点，车道，卡车，床头板，灯，叶子，照片，衣柜，雨伞，拱门，汽车，轮子，外套，电话，储物柜，辣椒，窗帘，工人，奶牛，露天平台，苹果，地球仪，油瓶，树，三角形，书，车库，商店，罐子，凳子，楼梯，门口，控制面板，消防栓，雕塑，跑步的人，太阳能板 |
| Dress, desk, vase, flower, wallpaper, chair, nail polish, table lamp, petal, bed, shirt, bottle, sofa, microwave, bag, glasses, boots, bicycle, action figure, helmet, plant, bow tie, blanket, tape, pillow, paper, tie, pendant, necklace, sunglasses, pants, house, roof, wine glass, beard, earrings, strawberry, life jacket, vending machine, mask, toy gun, display stand, microphone, computer keyboard, mouse, orange, cherry, bouquet, dice, coffee, ornament, penguin, pumpkin, goggles, bear, star, hair, uniform, washing machine, sweatshirt, shed, yard, banner, dessert, driveway, truck, headboard, lamp, leaf, photo, wardrobe, umbrella, arch, car, wheel, coat, telephone, locker, chili pepper, curtain, worker, cow, patio, apple, globe, oil bottle, tree, triangle, book, garage, store, jar, stool, staircase, doorway, control panel, fire hydrant, sculpture, running person, solar panel |

| id:101-200 |
| --- |
| 原木，地毯，小提琴，指甲，骆驼，篮球，跑道，厨师，碗，枝形吊灯，袍子，哑铃，雕像，冰块，烤箱，发带，餐具，郁金香，马，藤椅，戒指，托盘，杯子，购物袋，拖拉机，乌龟，纪念碑，甜甜圈，鹿，玩具，玩具车，行进乐队，女人，手提包，卧室，软管，手套，剪刀，叉子，桶，眼睛，托特包，笔记本电脑，缝纫机，光线，钢琴，跑车，滑梯，柠檬，摄像机，车头灯，蜜蜂，瓢虫，刀，披萨，蝴蝶，小雕像手办，人行步道，门，走廊，水族馆，场地，口袋，枪，宠物牵引绳，围栏，链条，钱包，兰花，轮胎，铅笔，耳机，柱子，长椅，灌木丛，赛车，花盆，葡萄，桨，木筏，岩石，番茄，柜台，指甲油瓶，鼓，玩具拖拉机，操作台，镜头，小鸟，婚宴场地，头巾，健身器械，茶包，灯泡，帽子，精灵，橡胶圈，黄油，立方体，棒球手套 |
| Log, carpet, violin, fingernail, camel, basketball, runway, chef, bowl, chandelier, robe, dumbbell, statue, ice cube, oven, hairband, cutlery, tulip, horse, rattan chair, ring, tray, cup, shopping bag, tractor, turtle, monument, donut, deer, toy, toy car, marching band, woman, handbag, bedroom, hose, gloves, scissors, fork, bucket, eye, tote bag, laptop, sewing machine, light ray, piano, sports car, slide, lemon, camera, headlight, bee, ladybug, knife, pizza, butterfly, figurine, sidewalk, door, corridor, aquarium, venue, pocket, gun, pet leash, fence, chain, wallet, orchid, tire, pencil, headphones, pillar, bench, bush, race car, flowerpot, grape, paddle, raft, rock, tomato, counter, nail polish bottle, drum, toy tractor, workbench, lens, bird, wedding venue, headscarf, fitness equipment, tea bag, light bulb, hat, elf, rubber band, butter, cube, baseball glove |

| id:201-300 |
| --- |
| 棒球棒，礼服，箭头，警戒带，纸杯蛋糕，手链，香水瓶，蔬菜，海报，双层床，奶酪，摩托车，吸尘器，猫头鹰，搅拌杯，牛奶，宝石，古董车，手推车，背心，多肉植物，轨道，火车头，汉堡，洗碗机，公寓楼群，海洋，轮椅，键盘，丝带，寿司卷，跑步机，书架，马尾辫，停车场，电池，扫帚，羊，压力表，谷仓，足球，浴缸，裙子，手杖，肉桂卷，旗帜，西瓜，西蓝花，水瓶，秋千床，休闲椅，鸡尾酒杯，火，曲棍球杆，珍珠，米饭，工艺品，腰带，淋浴间，爆米花，毛皮大衣，徽章，四柱床，骑自行车的人，栏杆，杆，鸽子，机器，码头，糕点，花坛，球体，湖泊，药片，编织袋，葡萄酒瓶，拼贴画，婴儿，奶酪块，眼影，雪花，浴室，向日葵，滑板车，公交车，茶壶，皮划艇，炉子，肥皂，毛巾，药丸，玻璃杯，香烟，站台，卷尺，充气椅，石头，工厂，毛衣，乳液 |
| Baseball bat, evening gown, arrow, caution tape, cupcake, bracelet, perfume bottle, vegetables, poster, bunk bed, cheese, motorcycle, vacuum cleaner, owl, mixing cup, milk, gemstone, vintage car, cart, vest, succulent plant, track, locomotive, hamburger, dishwasher, apartment complex, ocean, wheelchair, keyboard, ribbon, sushi roll, treadmill, bookshelf, ponytail, parking lot, battery, broom, sheep, pressure gauge, barn, soccer ball, bathtub, skirt, cane, cinnamon roll, flag, watermelon, broccoli, water bottle, swing bed, lounge chair, cocktail glass, fire, hockey stick, pearl, rice, craft, belt, shower stall, popcorn, fur coat, badge, four-poster bed, person riding a bicycle, railing, pole, pigeon, machine, dock, pastry, flower bed, sphere, lake, pill, woven bag, wine bottle, collage, baby, cheese block, eyeshadow, snowflake, bathroom, sunflower, scooter, bus, teapot, kayak, stove, soap, towel, pill, glass, cigarette, platform, tape measure, inflatable chair, stone, factory, sweater, lotion |

*Table G.2.* All categories in BoxClass-CN, grouped and displayed by ID (301-566).

| id:301-400 |
| --- |
| 后院, 人像模型, 火车, 毕业礼服, 猴子, 灯笼, 红酒架, 四轮越野车, 风车, 豆袋椅, 汤, 乐高人偶, 树桩, 圣诞树, 钳子, 绳子, 袖扣, 黑板, 连帽衫, 头冠, 眉毛, 报纸, 婴儿车, 婴儿连衣裙, 鸡尾酒, 毛巾架, 瑜伽垫, 狼, 山, 香槟, 埃菲尔铁塔, 阳台, 喷泉, 冠军腰带, 山羊, 面罩, 平底锅, 消防车, 爆米花机, 实验服, 风衣, 船坞, 信封, 啤酒瓶, 冲浪板, 面板, 塑料袋, 飞机, 窗台, 画笔, 洗漱包, 百叶窗, 婴儿床, 杏仁, 餐巾纸, 工具箱, 高尔夫球手, 香槟杯, 帆船, 干花, 行李, 瓷砖, 垃圾桶, 购物车, 梨, 密尔沃基工具包, 拖鞋, 山脉, 恐龙, 壁画, 高速公路, 梳子, 人字拖, 雪地靴, 士兵, 徒步旅行者, 叉车, 海绵, 燕尾服, 橄榄油, 齿轮, 加拿大国家电视塔, 小屋, 游戏机, 电线, 插头, 松鼠, 猕猴桃, 馅饼, 躺椅, 音乐家, 老虎, 沙滩椅, 咖啡袋, 网球拍, 吊灯, 蘑菇, 昆虫, 家庭办公室, 煎饼 |
| Backyard, mannequin, train, graduation gown, monkey, lantern, wine rack, ATV, windmill, bean bag chair, soup, Lego minifigure, tree stump, Christmas tree, pliers, rope, cufflinks, blackboard, hoodie, crown, eyebrow, newspaper, stroller, baby dress, cocktail, towel rack, yoga mat, wolf, mountain, champagne, Eiffel Tower, balcony, fountain, championship belt, goat, face mask, frying pan, fire truck, popcorn machine, lab coat, trench coat, boat dock, envelope, beer bottle, surfboard, panel, plastic bag, airplane, windowsill, paintbrush, toiletry bag, blinds, crib, almond, napkin, toolbox, golfer, champagne flute, sailboat, dried flowers, luggage, tile, trash can, shopping cart, pear, Milwaukee tool kit, slippers, mountain range, dinosaur, mural, highway, comb, flip-flops, snow boots, soldier, hiker, forklift, sponge, tuxedo, olive oil, gear, CN Tower, cabin, game console, wire, plug, squirrel, kiwi, pie, recliner, musician, tiger, beach chair, coffee bag, tennis racket, chandelier, mushroom, insect, home office, pancake |

| id:401-500 |
| --- |
| 编织吊椅, 玩具轨道, 花环, 茶, 钩针编织的苹果, 柠檬水, 火车车厢, 交通信号灯, 落地灯, 放大镜, 钩针编织花, 旋钮, 摇椅, 拳击袋, 眼线笔, 吧台凳, 棒球, 吉普车, 雨靴, 茎, 油量表, 柠檬角, 钢笔, 飞盘, 灭火器, 轮胎压力计, 面条, 长颈鹿, 煎锅, 菠萝, 加油站, 酒店, 香蕉, 拳击台, 车顶行李架, 纸巾, 电脑鼠标, 围巾, 考拉熊, 北极熊, 防水包, 3D眼镜, 婚纱, 甜椒, 刷子, 游乐场, 拳击手套, 音乐键盘, 泥塑人偶, 防毒面具, 电影, 船, 海星, 医用口罩, 杯子架, 滑雪靴, 化妆包, 香槟瓶, 毛绒玩具, 老虎机, 狮子, 海岸, 碗碟架, 发电站, 鞋架, 越野摩托车, 滑雪板, 起重船, 素描, 花园水管, 牛仔靴, 汽车发动机, 棒球戒指, 锅架, 胶带卷, 校车, 高架床, 衣架, 双层巴士, 电视屏幕, 烤面包机专用袋, 汽车脚垫, 金属花, 警车, 医疗包, 油, 玩具火车, 床单, 急救包, 餐垫, 旅行袋, 乐高汽车, 高脚椅, 高尔夫球车, 精油, 肉桂棒, 烘焙用冷却架, 雏菊, 垃圾袋, 机油 |
| Woven hanging chair, toy track, wreath, tea, crocheted apple, lemonade, train car, traffic light, floor lamp, magnifying glass, crocheted flower, knob, rocking chair, punching bag, eyeliner, bar stool, baseball, jeep, rain boots, stem, fuel gauge, lemon wedge, pen, frisbee, fire extinguisher, tire pressure gauge, noodles, giraffe, frying pan, pineapple, gas station, hotel, banana, boxing ring, roof rack, tissue, computer mouse, scarf, koala, polar bear, waterproof bag, 3D glasses, wedding dress, bell pepper, brush, playground, boxing gloves, music keyboard, clay figurine, gas mask, movie, boat, starfish, medical mask, cup holder, ski boots, makeup bag, champagne bottle, plush toy, slot machine, lion, coastline, dish rack, power plant, shoe rack, dirt bike, snowboard, crane ship, sketch, garden hose, cowboy boots, car engine, baseball ring, pot rack, roll of tape, school bus, loft bed, clothes hanger, double-decker bus, TV screen, toaster bag, car floor mat, metal flower, police car, medical kit, oil, toy train, bedsheet, first aid kit, placemat, travel bag, Lego car, high chair, golf cart, essential oil, cinnamon stick, baking cooling rack, daisy, garbage bag, motor oil |

| id:501-566 |
| --- |
| 玩具屋, 维京船, 蛋卷, 高性能经典跑车, 玩具键盘, 香水, 砂轮, 正装衬衫, 独木舟, 马球衬衫, 风扇, 啤酒杯, 雕刻人像, 录像带, 浴帘, 浴室防滑垫, 喷枪, 威士忌瓶, 方向盘, 耳机线, 刹车灯, 自动提款机, 拐杖, 战术腰带, 厕纸卷, 滑雪杖, 身体按摩油, 折叠椅, 奶瓶, 充电站, 狗窝, 存钱罐, 火车站, 面包卷, 冰淇淋机, 公文包, 旅行包, 蔬菜卷, 面部精油, 乳液瓶, 沙滩巾, 金属板, 烈酒杯, 邮差包, 电视摄像机, 猫窝, 杏仁糖花, 茶叶, 洗衣袋, 午餐包, 方糖, 眉笔, 威士忌杯, 芹菜条, 牙科椅, 电源线, 咖啡机, 沙发床, 温度计, 睡袋, 护肤油, 传送带, 西尔斯大厦, 曲棍球手套, 食品杂货袋, 卡车货厢 |
| Dollhouse, Viking ship, ice cream cone, high-performance classic sports car, toy keyboard, perfume, grinding wheel, dress shirt, canoe, polo shirt, fan, beer mug, carved figurine, videotape, shower curtain, bathroom non-slip mat, airbrush, whiskey bottle, steering wheel, headphone cable, brake light, ATM, crutch, tactical belt, toilet paper roll, ski pole, body massage oil, folding chair, baby bottle, charging station, doghouse, piggy bank, train station, bread roll, ice cream machine, briefcase, travel bag, vegetable roll, facial essential oil, lotion bottle, beach towel, metal plate, shot glass, messenger bag, TV camera, cat bed, marzipan flower, tea leaves, laundry bag, lunch bag, sugar cube, eyebrow pencil, whiskey glass, celery stick, dental chair, power cord, coffee machine, sofa bed, thermometer, sleeping bag, skincare oil, conveyor belt, Sears Tower, hockey glove, grocery bag, truck cargo bed |

