# OpenReview forum: "FG-CLIP 2: A Bilingual Fine-grained Vision-Language Alignment Model"
_ICML.cc/2026/Conference — ICML 2026 regular_

### Official Review · Reviewer_BtY6 · 2026-03-11

**Soundness:** 3
**Presentation:** 3
**Significance:** 3
**Originality:** 3
**Overall Recommendation:** 4
**Confidence:** 4

**Summary:**

This paper examines the theme of bilingual (English and Chinese) fine-grained vision-language alignment. The paper's primary contribution concerns the development of FG-CLIP 2, a novel two-stage framework that significantly improves upon existing models in both global semantic alignment and localized, fine-grained cross-modal understanding. To achieve this, the authors introduce a dual-caption strategy in the first stage and multiple discriminative objectives in the second stage, most notably a newly proposed Textual Intra-modal Contrastive (TIC) loss that helps the text encoder separate semantically similar but distinct descriptions. Furthermore, to address the scarcity of fine-grained evaluation datasets in Chinese, the authors contribute a comprehensive new benchmark suite, including long-caption retrieval (LIT-CN, DCI-CN, DOCCI-CN) and region-based classification (BoxClass-CN) datasets. Extensive experiments demonstrate that FG-CLIP 2 achieves state-of-the-art results across 29 datasets and 8 downstream tasks.

**Compliance With Llm Reviewing Policy:**

Affirmed.

**Key Questions For Authors:**

1. Regarding the TIC loss: The similarity threshold for filtering true positives is set to 0.95. How sensitive is the model's performance to this specific hyperparameter? Did you observe false negatives being pushed apart when setting this threshold lower?

2. In Stage 1, you utilize Qwen2.5-VL to generate context-aware descriptions. Have you experimented with different LMMs for caption generation, and how sensitive is the resulting global alignment to the hallucination rate or descriptive style of the chosen LMM?

3. Could you provide an estimate of the total computational cost (e.g., GPU/NPU hours) required for Stage I and Stage II? This context is important for researchers looking to build upon your methodology.

**Limitations:**

The authors honestly acknowledge their limitations in the conclusion. The primary limitations include:

1. Linguistic Scope: Despite its success in a bilingual setting, scaling this fine-grained alignment approach to a massive multilingual setting (>50 languages) might face exponential hurdles in data curation (especially obtaining high-quality region-text pairs and long captions).

2. Relational Understanding: The model currently lacks explicit structural modeling for relationships among objects (e.g., scene graph integration), which leaves room for failure cases in highly complex, multi-object interactive scenes.

3. Input Constraints: The maximum context length is extended to 196 tokens, which is an improvement, but may still fall short for ultra-long narrative descriptions of extremely dense images.

**Strengths And Weaknesses:**

Strengths:

- The proposed Textual Intra-modal Contrastive (TIC) loss is conceptually simple yet highly effective. By pushing apart top-k hard negatives strictly within the text modality, the model elegantly solves the issue of text encoders lacking discriminative pressure for fine-grained grounding.

- The introduction of the Chinese fine-grained multimodal benchmarks (e.g., BoxClass-CN, LIT-CN) is a highly valuable contribution to the community. It fills a critical gap, as most existing fine-grained benchmarks are heavily skewed towards English.

- The model shows exceptional versatility and robustness. It not only achieves SOTA on standard retrieval and fine-grained classification tasks but also proves to be a superior plug-and-play vision encoder for open-vocabulary detection (with LLMDet), dense segmentation (with Cat-Seg), and Large Multimodal Models.

Weaknesses:

- In Stage 1, the model uses LMM-generated long captions to augment short captions. However, there is no analysis or ablation regarding how the quality or length of these generated captions affects the final alignment performance.

- While fine-grained understanding is heavily evaluated (attributes, colors, etc.), the paper lacks a deep dive into explicit compositional and relational reasoning (e.g., spatial relations like "A is to the left of B but not C").

---

> ### Author Rebuttal · Authors · 2026-03-31
>
> **Response to W1 and Q2 (Caption quality, length, and LMM sensitivity)**
>
> Thank you for these important questions. We agree that caption quality and generation style are relevant to Stage I supervision. A full cross-LMM re-generation study would be ideal, but re-generating our 2.5B-scale corpus with multiple LMMs is not feasible within the rebuttal timeline. To directly probe the same concerns in a controlled way, we conducted two targeted stress tests. For efficiency, these additional experiments were performed on the English-only training data, rather than the full bilingual corpus.
>
> First, for caption-length sensitivity, we shortened all Stage I and Stage II captions to approximately 50% of their original length (by sentence truncation at punctuation boundaries). This tests whether long-form supervision provides additional value beyond shorter descriptions. Second, for hallucination/style sensitivity, we introduced controlled caption corruption by selecting 30% of training samples and replacing 30% of caption content with mismatched content from other samples, simulating LMM hallucination-like noise and style inconsistency.
>
> |Setting|DCI I→T|DCI T→I|COCO I→T|COCO T→I|FG-H|FG-M|FG-E|FG-Tr|
> |:-|:-:|:-:|:-:|:-:|:-:|:-:|:-:|:-:|
> |**Baseline**|**63.4**|**64.1**|**73.0**|**54.4**|**51.2**|**76.6**|**78.3**|**88.1**|
> |50% trunc.|61.7|61.5|72.0|54.1|49.5|74.6|77.5|87.8|
> |Δ trunc.|−1.7|−2.6|−1.0|−0.3|−1.7|−2.0|−0.8|−0.3|
> |30%×30% halluc.|62.3|62.7|72.8|53.9|49.4|73.9|75.9|87.8|
> |Δ halluc.|−1.1|−1.4|−0.2|−0.5|−1.8|−2.6|−2.4|−0.3|
>
> Key findings:
> - Standard retrieval (MSCOCO) is robust to both perturbations (Δ ≤ 1 pp).
> - Long-text retrieval (DCI) is sensitive to both, as dense visual matching relies on complete and accurate details.
> - Fine-grained understanding (FG-OVD) is more sensitive to hallucination than truncation. For example, FG-Medium/Easy drops are −2.6/−2.4 under hallucination vs −2.0/−0.8 under truncation.
>
> This confirms that wrong information is more harmful than missing information: truncation removes details but preserves correctness, while hallucination corrupts attribute-level decision boundaries.
>
> In addition, we performed small-batch human evaluation across multiple LMM/prompt settings before large-scale caption generation, and iteratively refined prompts to reduce hallucination and improve caption fidelity. Therefore, the Stage-I captions were quality-controlled before full-scale generation.
>
> **Response to W2**
>
> Thank you for this suggestion. We have addressed this by evaluating FG-CLIP 2 on four standard compositional and relational reasoning benchmarks: Winoground, ARO, SugarCrepe, and SugarCrepe++. Please refer to the **Response to Q3 for Reviewer D4Nf** for detailed results and analysis.
>
> **Response to Q1**
>
> Thank you for this question. To examine the sensitivity of the 0.95 threshold, we conducted additional experiments with thresholds of 0.85 and 0.75. We visualize the similarity matrices under different thresholds in https://anonymous.4open.science/r/77hyujt/show.md.
>
> The visualizations show that when the threshold is reduced, more semantically close captions are filtered out from the hard-negative set, which weakens fine-grained contrastive supervision. At the same time, the positive caption is less stably ranked as the top match, and the margin to nearby captions becomes smaller (with reversals in some cases). In this sense, lowering the threshold can also be viewed as increasing false-negative effects at ranking time: true positives are more easily confused with highly similar captions.
>
> Therefore, a lower threshold both reduces informative hard-negative learning and increases confusion among nearby captions, making fine-grained discrimination harder. The 0.95 setting provides a better balance between filtering near-duplicate positives and preserving useful contrastive signals.
>
> **Response to Q3**
>
> Thank you for this question. We agree that reporting training cost is important for reproducibility. We summarize the total computational cost of Stage I and Stage II below.
>
> | Stage | Hardware | Devices | Model | Wall-clock (h) | Device-Hours |
> | --- | --- | --- | --- | --- | --- |
> | Stage I | Ascend 910B NPU | 160 | ViT-B | 88 | 14,080 NPU-h |
> | Stage I | Ascend 910B NPU | 160 | ViT-L | 175 | 28,000 NPU-h |
> | Stage I | Ascend 910B NPU | 160 | ViT-So | 254 | 40,640 NPU-h |
> | Stage II | NVIDIA H800 GPU | 16 | ViT-B | 8 | 128 GPU-h |
> | Stage II | NVIDIA H800 GPU | 16 | ViT-L | 16 | 256 GPU-h |
> | Stage II | NVIDIA H800 GPU | 16 | ViT-So | 24 | 384 GPU-h |

---

### Official Review · Reviewer_jbMu · 2026-03-11

**Soundness:** 4
**Presentation:** 3
**Significance:** 3
**Originality:** 2
**Overall Recommendation:** 5
**Confidence:** 5

**Summary:**

The paper presents a family of contrastive multimodal encoders. The proposed model are trained with combination of various advanced techniques and methods, and eventually achieved compitive performance across benchmarks compared with state-of-the-art cross-model encoders. More importantly, the vision encoder empirically serve as strong encoder for building multimodal large language models, suggesting strong contribution of the model family.

**Compliance With Llm Reviewing Policy:**

Affirmed.

**Final Justification:**

The response regarding equation similarity to prior works and the motivation behind and the key differences in data synthesis resolve my primary concerns, and I acknowledge the successful training of huge models in such huge dataset with impressive results are good enought as contribution. The experiments on llava and other benchmarks further prove the effective training strategy proposed in the model, and scalling up training worth credit and therefore I improve the score to 5

**Key Questions For Authors:**

1. The proposed TIC seems the same as eq3 in [1] ?
2. How does the author justify that improvement performance come from better data quality instead of better training and learning algorithm?

Zhang L, Awal R, Agrawal A. Contrasting intra-modal and ranking cross-modal hard negatives to enhance visio-linguistic compositional understanding[C]//Proceedings of the IEEE/CVF Conference on Computer Vision and Pattern Recognition. 2024: 13774-13784.

**Limitations:**

no.

**Strengths And Weaknesses:**

# Strengths
The primary contribution of FG-CLIP 2 is the delivery of a high-performance, multi-scale model family (from ViT-B to ViT-So) that provides the community with powerful new options for bilingual vision-language tasks. It demonstrates superior performance across 29 datasets and 8 distinct tasks, proving its versatility as a robust foundation for downstream applications like object detection and multimodal reasoning. Furthermore, the introduction of new evaluation suites like LIT-CN and BoxClass-CN provides essential tools for assessing fine-grained Chinese multimodal understanding, effectively filling a gap in the existing benchmark landscape.

# Weakness
The model exhibits a lack of methodology originality, as nearly all of its components—including the dual-encoder architecture, sigmoid loss, and ranking objectives—are borrowed from existing works like SigLIP and CE-CLIP. Its success relies on the effective engineering and integration of these pre-existing methods rather than novel algorithmic innovation. But this would result in a technical report style rather than a research paper. Additionally, the model is heavily dependent on high-quality synthetic data generated by other large multimodal models.

---

> ### Author Rebuttal · Authors · 2026-03-31
>
> **Response to Weakness**
>
> Thank you for this important comment. We agree that FG-CLIP 2 builds on strong prior foundations such as the dual-encoder design and SigLIP-style global alignment. Rather than claiming a completely new architecture, our methodological advance lies in introducing targeted improvements and organizing them into a coherent two-stage training strategy for bilingual fine-grained alignment. This includes practical architectural adaptations, the Textual Intra-modal Contrastive (TIC) loss as a dynamic hard-negative text supervision objective, and region-level objectives integrated under a single training pipeline.
>
> Beyond the training methodology, we also contribute a systematic evaluation perspective that is currently underexplored for Chinese-English fine-grained alignment. This includes new Chinese long-caption image-text retrieval benchmarks (LIT-CN, DCI-CN, and DOCCI-CN) as well as a Chinese region-based classification benchmark (BoxClass-CN). In this sense, the work is not intended as a pure engineering report, but as a comprehensive study of how fine-grained bilingual supervision and objective design affect representation quality and downstream transfer.
>
> Regarding dependence on synthetic data, we agree that data quality matters. However, as shown by our controlled comparisons (including matched-data baselines) in Appendix E, data alone is not sufficient to reproduce the gains without the proposed framework and training methodology. We will clarify this point more explicitly in the final version.
>
> **Response to Q1**
>
> Thank you for pointing this out. We agree that our TIC loss and Eq. (3) in [1] share a similar contrastive form at the equation level. However, they are fundamentally different in the source of hard negatives (rule-based vs. data-driven), the granularity they operate on (global vs. region-level), and the strategy for handling boundary ambiguity.
>
> In [1], the loss is driven by predefined rule-based data. The negatives ($T_{rel},T_{att},T_{act},T_{obj}$) are generated through explicit modifications of relation, attribute, action, or object. The boundaries between these global captions are clear by design, and the objective essentially serves as a standard multi-class loss tailored for global compositional understanding.
>
> In contrast, our TIC is driven by data-driven fine-grained conceptual similarities. In our region-level setting, many region descriptions within a batch are semantically close but refer to distinct visual regions, leading to inherently ambiguous boundaries and representation crowding. To address this, TIC does not rely on predefined rules. Instead, it dynamically mines hard negatives from within a batch by calculating pairwise region-text similarities. Crucially, to handle the ambiguous boundaries between local captions, we introduce a dual-strategy design: filtering out pairs with similarity > 0.95 (to avoid over-penalization) and selecting only the top-10 most similar texts as hard negatives (to focus on the most confusing concepts).
>
> Therefore, while both share a contrastive form, [1] uses a standard loss driven by rule-generated data for global tasks, whereas our TIC is a specialized dual-design (data-driven mining + boundary-aware filtering) specifically developed to resolve local representation crowding in fine-grained alignment.
>
> **Response to Q2**
>
> Thank you for this important question. We agree that stronger data quality can improve performance, so separating data effects from algorithmic effects is necessary.
>
> In Appendix E (Table D), we already provide a fully controlled comparison under identical data conditions. We retrained SigLIP 2 and Chinese-CLIP using exactly the same Stage II training data as FG-CLIP 2, and importantly, FG-CLIP 2 also uses only Stage II data in this comparison (no Stage I pretraining for any method). Under this setting, FG-CLIP 2 consistently outperforms both baselines across all reported benchmarks, including DCI, MSCOCO, DOCCI-CN, Flickr30k-CNA, COCO⁸⁰, and Fine-Grained Understanding, with substantial margins. This shows that the gains cannot be explained by data quality alone.
>
> To further strengthen this point, we additionally trained SigLIP 2 with the complete Stage I + Stage II data pipeline. Please refer to the **Response to W1 for Reviewer mXVC** for detailed results. Under this fully matched setup, the two models are comparable on standard retrieval benchmarks, while FG-CLIP 2 still performs substantially better on long-text retrieval, bbox classification, and fine-grained understanding.
>
> Taken together, these matched-data comparisons indicate that data quality alone cannot explain the gains, and that our training and learning design plays a key role in the final performance.
>
> **Response to Limitations**
>
> Our limitations are implicitly reflected in our "future work" statement (e.g., extending to longer textual inputs and broader settings). We will make this point more explicit in the last section.

---

> > ### Author Rebuttal · Reviewer_jbMu · 2026-04-02
> >
> > The response solve my prior concerns and questions. I agree that while from the algorithm perspective the equation is very similar, however the data synthesis strategy and motivation might differ in different settings. And the experimental results are strong enough as a contribution to the community.

---

> > > ### Author Response · Authors · 2026-04-03
> > >
> > > Dear Reviewer jbMu,
> > >
> > > Thank you very much for your time, your constructive feedback, and for acknowledging our rebuttal. We are glad that our response has resolved your prior concerns and that you find our experimental results and contributions valuable to the community.
> > >
> > > We will make sure to incorporate these clarifications into the final version of the paper to further improve its clarity.
> > >
> > > Thank you again for your support!

---

### Official Review · Reviewer_D4Nf · 2026-03-12

**Soundness:** 3
**Presentation:** 3
**Significance:** 2
**Originality:** 2
**Overall Recommendation:** 3
**Confidence:** 4

**Summary:**

The authors present FG-CLIP 2, a vision language model combining both bilingual English/Chinese capabilities and fine-grained multimodal understanding.
Building upon SigLIP 2, the authors propose a 2-stage fine-tuning strategy.
Stage I establishes coarse-to-fine global alignment using both short and long captions, combining data from LAION-2B (long captions generated with an LMM) and a combined 750M Chinese dataset. Stage II instead uses datasets (FineHARD + a corresponding Chinese in-house dataset) with region-level information defined by bounding boxes and corresponding textual descriptions and hard negatives, while employing 4 additional loss functions, 1 of which, the Textual Intra-modal Contrastive (TIC) loss, is novel. The authors also contribute new Chinese evaluation benchmarks: three long-caption retrieval datasets (LIT-CN, DCI-CN, DOCCI-CN) and a region-classification benchmark (BoxClass-CN with 566 categories). Experiments across 29 datasets and 8 tasks demonstrate consistent improvements over strong baselines, including SigLIP 2, MetaCLIP 2, and FG-CLIP for both languages.

**Compliance With Llm Reviewing Policy:**

Affirmed.

**Final Justification:**

Despite the authors’ rebuttal satisfactorily addressing most of my technical questions and clarifying the paper’s motivations, I remain unconvinced about the overall significance of the contribution. In my view, the proposed method is still largely a combination of existing components and training strategies, with only limited methodological novelty beyond their integration and scaling. While the empirical results are strong and the new benchmarks are useful, I am not persuaded that this is the kind of contribution that will have broad impact on the community beyond advancing state of the art on a benchmark.

My main remaining concern is the reliance on an underspecified in-house dataset, which plays an important role in the reported gains. Even with the authors’ commitment to release the data, the current version leaves too much uncertainty about reproducibility and about how much of the improvement comes from the method versus the private data. For these reasons, although the paper has clear strengths, I continue to lean negative, as I believe these concerns outweigh the strengths at this stage.

**Key Questions For Authors:**

Please refer to weaknesses for major concerns, however I have also some additional questions:
- Table C shows that adding Chinese Stage II data improves English benchmarks. Can the authors provide an analysis or hypothesis for why this cross-lingual transfer occurs?
- What is the reasoning behind using different list of values when tuning the losses' weights in Figure B?
- The fine-grained supervision in FG-CLIP 2 is largely grounded in bounding-box-defined regions, primarily training the model to discriminate differences within cropped local patches. However, several important fine-grained distinctions are inherently relational and cannot be resolved by inspecting any single region in isolation. The paper evaluates 29 datasets but does not appear to include any standard whole-image compositional reasoning benchmarks such as Winoground, ARO, SugarCrepe, SugarCrepe++. How does FG-CLIP 2 perform on such benchmarks? Is there a risk that region-centric training improves local discrimination at the expense of global relational understanding?

**Limitations:**

Impact statement is correctly written and placed. However, Limitations are not discussed fairly, even if the last section is named "Conclusion and Limitations". That section mostly recaps the paper contribution and improvements, not discussing limitations honestly. I would suggest to include fair limitations discussion in that last section.

**Strengths And Weaknesses:**

Strengths
- The technical approach is well-motivated and clearly described. The two-stage training pipeline systematically addresses the known limitations of global contrastive pre-training. The TIC loss naturally includes intra-modal negatives, and the ablation in Table 8 demonstrates its contribution.
- The experimental coverage is vast. With 29 datasets spanning localization (FG-OVD, bounding-box classification, open-vocabulary detection), image-level retrieval (short and long text, multilingual), zero-shot classification, dense segmentation, and large multimodal model (LMM) integration, the paper provides a thorough validation across diverse downstream tasks.
- Bilingual Benchmark Contribution: The introduction of LIT-CN, DCI-CN, DOCCI-CN, and BoxClass-CN are useful benchmarks for evaluating Chinese multimodal models, enabling evaluation at a finer granularity. Native-speaker validation of the DCI-CN and DOCCI-CN translations adds credibility to LMM-generated translation and corrsepondence to images.

Weaknesses
- While the TIC loss is a useful contribution, the overall framework is a relatively straightforward combination of existing techniques (SigLIP 2 backbone, CMR loss from [Zhang et al. 2024b], hard negatives from FineHARD, region-level features, hard negatives and LAION-2B long captions from FG-CLIP). The primary novelty lies in combining these components bilingually and at scale.
- Stage I uses a 500M short/long caption-image pairs, while Stage II uses 12 million Chinese region-text pairs, both from an undisclosed in-house dataset. This significantly limits reproducibility. While the models are said to be made public, if the undisclosed data is playing a big role in the improvement, keeping it private would limit reproducibility and impact. The controlled comparison in Appendix E (retrained baselines on Stage II data only, without Stage I) partially addresses this aspect, but the full pipeline’s bilingual gains cannot be independently verified without most of the Chinese Language training data.

---

> ### Author Rebuttal · Authors · 2026-03-31
>
> **Response to W1**
>
> We agree FG-CLIP 2 builds on strong foundations, but our contribution is a systematic methodological package, not just a combination. At the objective level, we introduce the TIC loss to dynamically improve region-text separability (removing it drops COCO Top-1 from 74.9 to 70.1), along with practical architectural adaptations for fine-grained alignment.
>
> More broadly, we target the underexplored setting of bilingual (Chinese-English) fine-grained alignment and contribute novel Chinese benchmarks (LIT-CN, DCI-CN, DOCCI-CN, BoxClass-CN). Crucially, we provide extensive empirical analysis: the framework is validated across 29 datasets and 8 tasks, including thorough ablation studies on each objective, bilingual data composition effects, and matched-data comparisons. In particular, under matched-data retraining (Table D), FG-CLIP 2 still outperforms SigLIP 2 and Chinese-CLIP, proving the training methodology provides independent value beyond data scale.
>
> **Response to W2**
>
> To support reproducibility, we release the code, model weights, and 4 evaluation datasets. In addition, to directly address the reviewer's concern, we plan to release the Chinese in-house training data in the final version.
>
> Regarding the verifiability of bilingual gains, we have already provided supporting evidence in the paper. Appendix D (Table C) verifies bilingual gains: from the same Stage I checkpoint, adding Chinese Stage II data improves both Chinese and English benchmarks. We further trained an English-only full pipeline (removing Chinese data from Stage I+II), which performs worse than the bilingual setting (table below), further supporting bilingual contribution and reproducibility.
>
> |Method|DCI I→T/T→I|MSCOCO I→T/T→I|DOCCI-CN I→T/T→I|Flickr30k-CNA I→T/T→I|COCO⁸⁰ Top-1/Top-5|Fine-Grained Understanding Hard/Medium/Easy/Trivial|
> |:-|:-:|:-:|:-:|:-:|:-:|:-:|
> |FG-CLIP 2 (English-only in S1+S2)|63.4/64.1|**73.0**/54.4|46.2/46.8|78.8/56.4|67.7/94.4|51.2/**76.6**/78.3/88.1|
> |FG-CLIP 2|**64.5/64.9**|72.1/**54.5**|**71.2/75.4**|**85.4/69.9**|**74.9/95.7**|**52.3**/76.3/**80.3/92.0**|
>
> **Response to Q1**
>
> Thank you for this important point. We attribute the cross-lingual transfer in Table C to two main effects. First, since Chinese and English are aligned to the same visual encoder, adding Chinese Stage II data increases both global image-text and local region-text diversity, which enriches the overall semantic coverage and strengthens the shared representation space. Second, bilingual fine-grained training introduces richer lexical and compositional variation, which regularizes training and reduces overfitting to English-specific shortcuts. These explain why adding Chinese Stage II data yields measurable gains on English benchmarks.
>
> **Response to Q2**
>
> Our tuning protocol used a step size of 0.2 across all auxiliary losses, with loss-specific starting points determined by pilot sensitivity checks. For most losses (FGV, FGT, TIC), we searched [0.10, 0.30, 0.50, 0.70, 1.00]. For CMR, pilot runs showed that very small weights (around 0.10) produced weak ranking-margin effects, so we shifted the starting point to 0.20, yielding [0.20, 0.40, 0.60, 0.80, 1.00]. In all cases, when the best value on the coarse grid appeared near a boundary, we tested one additional finer point to verify whether the optimum lay at the boundary or beyond it. This is the reason why some candidate lists contain one extra value while others do not: the differences reflect boundary verification rather than inconsistent tuning rules.
>
> **Response to Q3**
>
> Thank you for this important concern. We evaluated on standard whole-image compositional benchmarks: Winoground, ARO, SugarCrepe, and SugarCrepe++. Each score below is the average across all sub-metrics within that benchmark.
>
> |Method|Winoground|ARO|SugarCrepe|SugarCrepe++|
> |:-|:-:|:-:|:-:|:-:|
> |SigLIP 2|20.33|34.62|78.79|62.40|
> |FG-CLIP 2 (w/o region-centric)|19.83|41.77|81.77|68.10|
> |FG-CLIP 2|**21.25**|**57.28**|**82.28**|**68.15**|
>
> Compared to SigLIP 2, FG-CLIP 2 improves on all four compositional benchmarks. Architecturally, we use a dedicated dense head to decouple region-level supervision from the global representation, which mitigates the risk of region-focused training interfering with global alignment. Nevertheless, to comprehensively evaluate the impact of region-centric training on compositional understanding, we trained a "w/o region-centric" ablation that removes all region-level objectives. The full model consistently outperforms this variant across all benchmarks, confirming that region-centric training does not come at the expense of global relational understanding but provides complementary benefits.
>
> **Response to Limitations**
>
> Thank you for the suggestion. Our limitations are implicitly reflected in our "future work" statement (e.g., extending to longer textual inputs and broader settings). We will make this point more explicit in the last section.

---

> > ### Author Rebuttal · Reviewer_D4Nf · 2026-04-02
> >
> > The authors addressed all concerns resolvable through rebuttal. I am still concerned about the use of undisclosed and underspecified in-house data, as it plays a big role and limits reproducibility in case of not being released in the future.

---

> > > ### Author Response · Authors · 2026-04-02
> > >
> > > Dear Reviewer D4Nf,
> > >
> > > Thank you very much for your time and for acknowledging that your all concerns have been adequately addressed.
> > >
> > > We understand your lingering concern regarding our in-house data. To clarify, this is not only a plan: we will release the Chinese in-house training data used in our work, together with the code, model weights, and the four evaluation datasets, in the final version. We have already prepared these resources for public release to ensure reproducibility.

---

### Official Review · Reviewer_mXVC · 2026-03-13

**Soundness:** 2
**Presentation:** 3
**Significance:** 2
**Originality:** 2
**Overall Recommendation:** 3
**Confidence:** 4

**Summary:**

The paper introduces FG-CLIP 2, a bilingual vision–language model designed to improve fine-grained cross-modal alignment between images and text in both English and Chinese. The paper's primary contribution concerns improving region-level and attribute-level alignment by augmenting a SigLIP-style dual-encoder architecture with additional supervision and objectives. The authors also introduce new Chinese multimodal benchmarks such as LIT-CN and BoxClass-CN.

**Compliance With Llm Reviewing Policy:**

Affirmed.

**Final Justification:**

My main concern was that the original comparison between SigLip2 and FG-CLIP2 might be unfair due to the additional training data used. The authors now provide results using the same training data, which mostly addresses this concern.

However, I believe SigLip2 trained on the same data should be clearly discussed and included as the baseline in all experimental tables, rather than showing only the original SigLip2. Therefore, I think the paper still needs some rewriting, and my rating remains the same.

**Key Questions For Authors:**

1. How would siglip2 perform with the same training data?
2. The performance on VQA tasks for LLava1.5+FG-CLIP2 seems moderate, can you show more fine-grained benchmarks results like V* and HSI-Bench?
3. How is Refcoco evaluated with LLava1.5+FG-CLIP2
4. Can you provide ablation on FGV and FGT losses

**Limitations:**

No. The authors can potentially discuss what is the impact on the model when training with multilingual datasets.

**Strengths And Weaknesses:**

Strengths

1. The work proposes a unified training framework that simultaneously improves fine-grained vision-language alignment and bilingual representation learning.
2. The paper introduces several new Chinese datasets which extend evaluation beyond short-caption retrieval to tasks such as long-caption retrieval and region-level classification.
3. The model is evaluated across 29 datasets covering 8 tasks.

Weaknesses

1. FG-CLIP 2 is built on SigLIP-2, but it is also trained with substantially additional supervision, including large bilingual datasets, long captions generated by LMMs, region-text pairs with bounding box annotations, synthetic hard negatives.
Because the baseline SigLIP-2 models used for comparison are not trained on these additional datasets, it is difficult to determine whether improvements arise from the proposed training objectives, or the additional training data.
Although the appendix includes some comparisons under shared data conditions, a clear baseline trained on the same dataset with only the original SigLIP loss would make the contribution easier to isolate. For example, the authors could start from stage 1 directly and train with either the proposed loss or only the siglip2 sigmoid loss. This would clarify whether gains stem from the proposed framework rather than data scale as the dataset is not part of the main contribution and also siglip2 performance improved significantly with additional data as shown in the appendix.

2. In Table 7, the authors evaluate LLaVA-1.5 + FG-CLIP-2 on RefCOCO. How is the evaluation done? Does the model predicts bounding boxes?  Also, given that the paper emphasizes fine-grained grounding ability, evaluating on fine-grained VQA datasets (e.g., V* or HSI-Bench) could further demonstrate the benefit of the proposed method.

3. The paper proposes multiple training objectives: FGV, FGT, TIC, CMR. It would be nice to also show ablation on FGV and FGT.

4. The Figure 1 is rather difficult to understand. It would be helpful to add explanation in the caption and also make annotations more informative.

---

> ### Author Rebuttal · Authors · 2026-03-31
>
> **Response to W1**
>
> Thank you for your important suggestion. We agree that isolating the effect of the proposed objectives from the effect of additional data is essential. Appendix E already includes shared-data comparisons, where retrained baselines use only Stage II data (without Stage I pretraining). These results indicate that data scale alone does not explain the gains.
>
> Following your suggestion, we further added a stricter controlled baseline: using the same Stage I + Stage II data pipeline as FG-CLIP 2, but training with only the siglip2 sigmoid loss. Under this fully matched setup, the two models are comparable on standard retrieval benchmarks, while FG-CLIP 2 still performs substantially better on long-text retrieval, bbox classification, and fine-grained understanding.
>
> |Setting|DCI I2T/T2I|MSCOCO I2T/T2I|DOCCI-CN I2T/T2I|Flickr30k-CNA I2T/T2I|COCO⁸⁰ Top1/Top5|FG Hard/Medium/Easy/Trivial|
> |:-|:-:|:-:|:-:|:-:|:-:|:-:|
> |SigLIP 2(original pretrained)|32.3/34.2|71.2/**55.2**|7.6/5.7|71.7/49.1|53.4/79.8|24.9/46.5/48.7/85.0|
> |SigLIP 2(Stage-II data)|49.1/49.0|71.5/53.5|50.5/49.4|81.0/52.7|53.7/79.7|25.5/50.6/59.2/83.4|
> |SigLIP 2(Stage-I+II data)|52.1/53.8|**72.3**/54.4|51.2/50.3|**86.3**/61.7|56.6/81.5|26.1/50.9/59.2/83.6|
> |FG-CLIP 2(Stage-II data)|60.9/62.6|72.1/53.7|65.9/69.5|84.3/66.1|71.0/95.4|51.9/76.0/**80.5**/90.6|
> |FG-CLIP 2(Stage-I+II data)|**64.5**/**64.9**|72.1/54.5|**71.2**/**75.4**|85.4/**69.9**|**74.9**/**95.7**|**52.3**/**76.3**/80.3/**92.0**|
>
> Taken together, the previous Stage-II-only shared-data comparison and the new fully matched Stage-I+II baseline provide consistent evidence that the improvements come from the proposed framework and fine-grained objectives, rather than from additional data alone. We will make this evidence more prominent in the final version.
>
> **Response to W2**
>
> For Table 7 (LLaVA-1.5 + FG-CLIP-2 on RefCOCO), we follow the original LLaVA-1.5 evaluation pipeline without introducing any extra region supervision or external region proposals at inference time. Specifically, we replace the original LLaVA visual encoder with the ViT of FG-CLIP 2, keep the rest of the LLaVA pipeline unchanged, and let LLaVA directly generate bounding box predictions in the same way as the original setting. Therefore, the RefCOCO results reflect the visual encoder replacement effect under a controlled evaluation protocol.
>
> Following your suggestion, we additionally evaluated on V* [1], a benchmark for fine-grained visual understanding. The results show consistent improvements after replacing the visual encoder with FG-CLIP 2, which is aligned with our claim that FG-CLIP 2 improves fine-grained representations. We will include these additional results in the final version.
>
> |V* Benchmark|Attribute Recognition|Spatial Relationship Reasoning|Overall|
> |-|-|-|-|
> |LLaVA-1.5 + CLIP|44.35|52.63|47.64|
> |LLaVA-1.5 + SigLIP 2|40.87|56.58|47.12|
> |LLaVA-1.5 + Meta CLIP 2|39.13|53.95|45.03|
> |LLaVA-1.5 + FG-CLIP 2|**45.22**|**57.89**|**50.26**|
>
> Regarding HSI-Bench [2], we found that this benchmark targets image-to-3D object reconstruction rather than our task setting, and it is currently not publicly available. Therefore, we do not consider it an appropriate or reproducible evaluation benchmark for the present paper.
>
> [1] Wu P, Xie S. V*: Guided visual search as a core mechanism in multimodal llms[C]//Proceedings of the IEEE/CVF Conference on Computer Vision and Pattern Recognition. 2024: 13084-13094
>
> [2] Cao Y, Xie H, Hong F, et al. HSImul3R: Reconstructing Simulation-Ready Human-Scene-Interaction from Sparse Views[J]. arXiv preprint arXiv:2603.15612, 2026
>
> **Response to W3**
>
> Thank you for this valuable suggestion. We added the requested ablations on FGV and FGT, shown below. Removing FGV causes a large drop on COCO⁸⁰ and consistent drops on fine-grained understanding, while removing FGT mainly hurts fine-grained understanding. This shows that FGV and FGT are complementary and both contribute to the full performance of FG-CLIP 2
>
> ||COCO⁸⁰-Top1|COCO⁸⁰-Top5|FG-H|FG-M|FG-E|FG-Tr|Flickr30k I→T|Flickr30k T→I|IN-V2|
> |-|-|-|-|-|-|-|-|-|-|
> |FG-CLIP 2|74.9|95.7|52.3|76.3|80.3|92.0|94.1|81.9|72.2|
> |w/o FGV|58.4|89.5|50.2|73.5|78.5|89.6|93.9|82.1|72.3|
> |w/o FGT|74.8|95.7|42.1|66.4|75.8|86.6|93.5|81.9|72.2|
>
> **Response to W4**
>
> Thank you for this helpful suggestion. We agree that Figure 1 can be made easier to follow. In the final version, we will improve both the caption and the in-figure annotations to make the architecture overview clearer and more informative.
>
> **Response to Limitations**
>
> We agree that the impact of multilingual training should be discussed more explicitly. In particular, while our English-Chinese setting shows mutually beneficial transfer, extending to broader multilingual settings may also introduce challenges such as cross-lingual interference, language imbalance, and representation-capacity bottlenecks. We will add this limitation discussion more clearly in the conclusion section.

---

> > ### Author Rebuttal · Reviewer_mXVC · 2026-04-05
> >
> > My main concern was that the original comparison between SigLip2 and FG-CLIP2 might be unfair due to the additional data used. The authors, however, provide new results using the same training data, which resolves this concern. I believe it is important that SigLip2 trained on the same data be included as the baseline in all experimental result tables, rather than showing only the original SigLip2.

---

> > > ### Author Response · Authors · 2026-04-05
> > >
> > > Dear Reviewer mXVC,
> > >
> > > We sincerely thank the reviewer for the constructive feedback and for acknowledging that the concerns have been fully addressed. We are glad that our additional experiments, specifically re-training SigLIP2 under the same data setting, help clarify the fairness of the comparison.
> > >
> > > We also appreciate your valuable suggestion regarding the presentation of results. Including SigLIP2 trained on the same data as a consistent baseline across all tables is important for clarity and completeness. We will incorporate these results into all experimental tables in the final version to further strengthen the paper.
> > >
> > > If possible, we would appreciate it if the reviewer could update the final justification to better reflect the current assessment of the work.

---

### Decision · Program_Chairs · 2026-04-30

**Decision:**

Accept (regular)

**Comment:**

The paper introduces FG-CLIP 2, a bilingual vision–language model designed to improve fine-grained cross-modal alignment between images and text in both English and Chinese. After rebuttal, it received mixed scores (3, 3, 4, 5). On the positive side, FG-CLIP 2 delivers a high-performance, multi-scale model family that provides the community with strong new baselines for bilingual vision–language tasks, demonstrating superior performance across 29 datasets and 8 task categories.

On the negative side, reviewers raised concerns about the level of novelty, as well as the fairness of comparisons with SigLip2, given the additional training data used for FG-CLIP 2. The authors have done a good job of rebuttal to address all these concerns, and all the reviewers who have participated in the discussion acknowledge that the concerns are fully resolved. Furthermore, one reviewer acknowledged that successfully training large-scale models on massive datasets, with consistently strong empirical results, constitutes a meaningful contribution in its own right. Overall, the AC thinks that the strengths of the paper outweigh its weaknesses and would like to recommend acceptance.